# Online and Differentially-Private Tensor Decomposition

**Yining Wang**
Machine Learning Department
Carnegie Mellon University
yiningwa@cs.cmu.edu

**Animashree Anandkumar**
Department of EECS
University of California, Irvine
a.anandkumar@uci.edu

## Abstract

Tensor decomposition is an important tool for big data analysis. In this paper, we resolve many of the key algorithmic questions regarding robustness, memory efficiency, and differential privacy of tensor decomposition. We propose simple variants of the tensor power method which enjoy these strong properties. We present the first guarantees for online tensor power method which has a linear memory requirement. Moreover, we present a noise calibrated tensor power method with efficient privacy guarantees. At the heart of all these guarantees lies a careful perturbation analysis derived in this paper which improves up on the existing results significantly.

**Keywords:** Tensor decomposition, tensor power method, online methods, streaming, differential privacy, perturbation analysis.

## 1 Introduction

In recent years, tensor decomposition has emerged as a powerful tool to solve many challenging problems in unsupervised [1], supervised [18] and reinforcement learning [4]. Tensors are higher order extensions of matrices which can reveal far greater information compared to matrices, while retaining most of the efficiencies of matrix operations [1].

A central task in tensor analysis is the process of decomposing the tensor into its rank-1 components, which is usually referred to as *CP (Candecomp/Parafac) decomposition* in the literature. While decomposition of arbitrary tensors is NP-hard [13], it becomes tractable for the class of tensors with linearly independent components. Through a simple *whitening* procedure, such tensors can be converted to orthogonally decomposable tensors. Tensor power method is a popular method for computing the decomposition of an orthogonal tensor. It is simple and efficient to implement, and a natural extension of the matrix power method.

In the absence of noise, the tensor power method correctly recovers the components under a random initialization followed by deflation. On the other hand, perturbation analysis of tensor power method is much more delicate compared to the matrix case. This is because the problem of tensor decomposition is NP-hard, and if a large amount of arbitrary noise is added to an orthogonal tensor, the decomposition can again become intractable. In [1], guaranteed recovery of components was proven under bounded noise, and the bound was improved in [2]. In this paper, we significantly improve upon the noise requirements, i.e. the extent of noise that can be withstood by the tensor power method.

In order for tensor methods to be deployed in large-scale systems, we require fast, parallelizable and scalable algorithms. To achieve this, we need to avoid the exponential increase in computation and memory requirements with the order of the tensor; i.e. a naive implementation on a 3rd-order $d$-dimensional tensor would require $O(d^3)$ computation and memory. Instead, we analyze the online tensor power method that requires only linear (in $d$) memory and does not form the entire tensor. This is achieved in settings, where the tensor is an empirical higher order moment, computed from the stream of data samples. We can avoid explicit construction of the tensor by running online tensor

power method directly on i.i.d. data samples. We show that this algorithm correctly recovers tensor components in time[1] $\tilde{O}(nk^2d)$ and $\tilde{O}(dk)$ memory for a rank-$k$ tensor and $n$ number of data samples. Additionally, we provide efficient sample complexity analysis.

As spectral methods become increasingly popular with recommendation system and health analytics applications [29, 17], data privacy is particularly relevant in the context of preserving sensitive private information. Differential privacy could still be useful even if data privacy is not the prime concern [30]. We propose the first differentially private tensor decomposition algorithm with both privacy and utility guarantees via noise calibrated power iterations. We show that under the natural assumption of tensor incoherence, privacy parameters have no (polynomial) dependence on tensor dimension $d$. On the other hand, straightforward input perturbation type methods lead to far worse bounds and do not yield guaranteed recovery for all values of privacy parameters.

## 1.1 Related work

**Online tensor SGD** Stochastic gradient descent (SGD) is an intuitive approach for online tensor decomposition and has been successful in practical large-scale tensor decomposition problems [16]. Despite its simplicity, theoretical properties are particularly hard to establish. [11] considered a variant of the SGD objective and proved its correctness. However, the approach in [11] only works for even-order tensors and its sample complexity dependency upon tensor dimension $d$ is poor.

**Tensor PCA** In the *statistical tensor PCA* [24] model a $d \times d \times d$ tensor $\mathbf{T} = \boldsymbol{v}^{\otimes 3} + \mathbf{E}$ is observed and one wishes to recover component $\boldsymbol{v}$ under the presence of Gaussian random noise $\mathbf{E}$. [24] shows that $\|\mathbf{E}\|_{\mathrm{op}} = O(d^{-1/2})$ is sufficient to guarantee approximate recovery of $\boldsymbol{v}$ and [14] further improves the noise condition to $\|\mathbf{E}\|_{\mathrm{op}} = O(d^{-1/4})$ via a 4th-order sum-of-squares relaxation. Techniques in both [24, 14] are rather complicated and could be difficult to adapt to memory or privacy constraints. Furthermore, in [24, 14] only one component is considered. On the other hand, [25] shows that $\|\mathbf{E}\|_{\mathrm{op}} = O(d^{-1/2})$ is sufficient for recovering multiple components from noisy tensors. However, [25] assumes exact computation of rank-1 tensor approximation, which is NP-hard in general.

**Noisy matrix power methods** Our relaxed noise condition analysis for tensor power method is inspired by recent analysis of noisy matrix power methods [12, 6]. Unlike the matrix case, tensor decomposition no longer requires *spectral gap* among eigenvalues and eigenvectors are usually recovered one at a time [1, 2]. This poses new challenges and requires non-trivial extensions of matrix power method analysis to the tensor case.

## 1.2 Notation and Preliminaries

We use $[n]$ to denote the set $\{1, 2, \cdots, n\}$. We use bold characters $\mathbf{A}, \mathbf{T}, \boldsymbol{v}$ for matrices, tensors, vectors and normal characters $\lambda, \mu$ for scalars. A $p$th order tensor $\mathbf{T}$ of dimensions $d_1, \cdots, d_p$ has $d_1 \times \cdots \times d_p$ elements, each indexed by a $p$-tuple $(i_1, \cdots, i_p) \in [d_1] \times \cdots \times [d_p]$. A tensor $\mathbf{T}$ of dimensions $d \times \cdots \times d$ is *super-symmetric* or simply *symmetric* if $\mathbf{T}_{i_1, \cdots, i_p} = \mathbf{T}_{\sigma(i_1), \cdots, \sigma(i_p)}$ for all permutations $\sigma : [p] \to [p]$. For a tensor $\mathbf{T} \in \mathbb{R}^{d_1 \times \cdots \times d_p}$ and matrices $\mathbf{A}_1 \in \mathbb{R}^{m_1 \times d_1}, \cdots, \mathbf{A}_p \in \mathbb{R}^{m_p \times d_p}$, the *multi-linear form* $\mathbf{T}(\mathbf{A}_1, \cdots, \mathbf{A}_p)$ is a $m_1 \times \cdots \times m_p$ tensor defined as

$$[\mathbf{T}(\mathbf{A}_1, \cdots, \mathbf{A}_p)]_{i_1, \cdots, i_p} = \sum_{j_1 \in [d_1]} \cdots \sum_{j_p \in [d_p]} \mathbf{T}_{j_1, \cdots, j_p} [\mathbf{A}_1]_{j_1, i_1} \cdots [\mathbf{A}_p]_{j_p, i_p}.$$

We use $\|\boldsymbol{v}\|_2 = \sqrt{\sum_i \boldsymbol{v}_i^2}$ for vector 2-norm and $\|\boldsymbol{v}\|_\infty = \max_i |\boldsymbol{v}_i|$ for vector infinity norm. We use $\|\mathbf{T}\|_{\mathrm{op}}$ to denote the *operator norm* or *spectral norm* of a tensor $\mathbf{T}$, which is defined as $\|\mathbf{T}\|_{\mathrm{op}} = \sup_{\|\boldsymbol{u}_1\|_2 = \cdots = \|\boldsymbol{u}_p\|_2 = 1} \mathbf{T}(\boldsymbol{u}_1, \cdots, \boldsymbol{u}_p)$. An event $\mathcal{A}$ is said to occur *with overwhelming probability* if $\Pr[\mathcal{A}] \geq 1 - d^{-10}$.

We limit ourselves to symmetric 3rd-order tensors ($p = 3$) in this paper. The results can be directly extended to asymmetric tensors since they can first be symmetrized using simple matrix operations (see [1]). Extension to higher-order tensors is also straightforward. A symmetric 3rd-order tensor $\mathbf{T}$ is rank-1 if it can be written in the form of

$$\mathbf{T} = \lambda \cdot \boldsymbol{v} \otimes \boldsymbol{v} \otimes \boldsymbol{v} = \lambda \boldsymbol{v}^{\otimes 3} \quad \Longleftrightarrow \quad \mathbf{T}_{i,j,\ell} = \lambda \cdot \boldsymbol{v}(i) \cdot \boldsymbol{v}(j) \cdot \boldsymbol{v}(\ell), \tag{1}$$

**Algorithm 1** Robust tensor power method [1]

---

1: **Input**: symmetric $d \times d \times d$ tensor $\widetilde{\mathbf{T}}$, number of components $k \leq d$, number of iterations $L, R$.
2: **for** $i = 1$ to $k$ **do**
3:     *Initialization*: Draw $\boldsymbol{u}_0$ uniformly at random from the unit sphere in $\mathbb{R}^d$.
4:     *Power iteration*: Compute $\boldsymbol{u}_t = \widetilde{\mathbf{T}}(\mathbf{I}, \boldsymbol{u}_{t-1}, \boldsymbol{u}_{t-1})/\|\widetilde{\mathbf{T}}(\mathbf{I}, \boldsymbol{u}_{t-1}, \boldsymbol{u}_{t-1})\|_2$ for $t = 1, \cdots, R$.
5:     *Boosting*: Repeat Steps 3 and 4 for $L$ times and obtain $\boldsymbol{u}_R^{(1)}, \cdots, \boldsymbol{u}_R^{(L)}$. Let $\tau^* = \operatorname{argmax}_{\tau=1}^{L} \widetilde{\mathbf{T}}(\boldsymbol{u}_R^{(\tau)}, \boldsymbol{u}_R^{(\tau)}, \boldsymbol{u}_R^{(\tau)})$. Set $\hat{v}_i = \boldsymbol{u}_R^{(\tau)}$ and $\hat{\lambda}_i = \widetilde{\mathbf{T}}(\boldsymbol{u}_R^{(\tau)}, \boldsymbol{u}_R^{(\tau)}, \boldsymbol{u}_R^{(\tau)})$.
6:     *Deflation*: $\widetilde{\mathbf{T}} \leftarrow \widetilde{\mathbf{T}} - \hat{\lambda}_i \hat{v}_i^{\otimes 3}$.
7: **end for**
8: **Output**: Estimated eigenvalue/Eigenvector pairs $\{\hat{\lambda}_i, \hat{v}_i\}_{i=1}^{k}$.

---

where $\otimes$ represents the *outer product*, and $\boldsymbol{v} \in \mathbb{R}^d$ is a unit vector (i.e., $\|\boldsymbol{v}\|_2 = 1$) and $\lambda \in \mathbb{R}^+$. [2] A tensor $\mathbf{T} \in \mathbb{R}^{d \times d \times d}$ is said to have a CP (Candecomp/Parafac) *rank k* if it can be (minimally) written as the sum of $k$ rank-1 tensors:

$$\mathbf{T} = \sum_{i \in [k]} \lambda_i \boldsymbol{v}_i \otimes \boldsymbol{v}_i \otimes \boldsymbol{v}_i, \quad \lambda_i \in \mathbb{R}^+, \ \ \boldsymbol{v}_i \in \mathbb{R}^d. \tag{2}$$

A tensor is said to be orthogonally decomposable if in the above decomposition $\langle \boldsymbol{v}_i, \boldsymbol{v}_j \rangle = 0$ for $i \neq j$. Any tensor can be converted to an orthogonal tensor through an invertible whitening transform, provided that $\boldsymbol{v}_1, \boldsymbol{v}_2, \ldots, \boldsymbol{v}_k$ are linearly independent [1]. We thus limit our analysis to orthogonal tensors in this paper since it can be extended to this more general class in a straightforward manner.

**Tensor Power Method:** A popular algorithm for finding the tensor decomposition in (2) is through the tensor power method. The full algorithm is given in Algorithm 1. We first provide an improved noise analysis for the robust power method, improving error tolerance bounds previously established in [1]. We next propose memory-efficient and/or differentially private variants of the robust power method and give performance guarantee based on our improved noise analysis.

## 2 Improved Noise Analysis for Tensor Power Method

When the tensor $\mathbf{T}$ has an exact orthogonal decomposition, the power method provably recovers all the components with random initialization and deflation. However, the analysis is more subtle under noise. While matrix perturbation bounds are well understood, it is an open problem in the case of tensors. This is because the problem of tensor decomposition is NP-hard, and becomes tractable only under special conditions such as orthogonality (and more generally linear independence). If a large amount of arbitrary noise is added, the decomposition can again become intractable. In [1], guaranteed recovery of components was proven under bounded noise and we recap the result below.

**Theorem 2.1** ([1] Theorem 5.1, simplified version). *Suppose* $\widetilde{\mathbf{T}} = \mathbf{T} + \boldsymbol{\Delta}_T$, *where* $\mathbf{T} = \sum_{i=1}^{k} \lambda_i \boldsymbol{v}_i^{\otimes 3}$ *with* $\lambda_i > 0$ *and orthonormal basis vectors* $\{\boldsymbol{v}_1, \cdots, \boldsymbol{v}_k\} \subseteq \mathbb{R}^d$, $d \geq k$, *and noise* $\boldsymbol{\Delta}_T$ *satisfies* $\|\boldsymbol{\Delta}_T\|_{\mathrm{op}} \leq \epsilon$. *Let* $\lambda_{\max}, \lambda_{\min}$ *be the largest and smallest values in* $\{\lambda_i\}_{i=1}^{k}$ *and* $\{\hat{\lambda}_i, \hat{v}_i\}_{i=1}^{k}$ *be outputs of Algorithm 1. There exist absolute constants* $K_0, C_1, C_2, C_3 > 0$ *such that if*

$$\epsilon \leq C_1 \cdot \lambda_{\min}/d, \quad R = \Omega(\log d + \log \log(\lambda_{\max}/\epsilon)), \quad L = \Omega(\max\{K_0, k\} \log(\max\{K_0, k\})), \tag{3}$$

*then with probability at least* 0.9, *there exists a permutation* $\pi : [k] \to [k]$ *such that*

$$|\lambda_i - \hat{\lambda}_{\pi(i)}| \leq C_2 \epsilon, \quad \|\boldsymbol{v}_i - \hat{\boldsymbol{v}}_{\pi(i)}\|_2 \leq C_3 \epsilon/\lambda_i, \quad \forall i = 1, \cdots, k.$$

Theorem 2.1 is the first provably correct result on robust tensor decomposition under general noise conditions. In particular, the noise term $\boldsymbol{\Delta}_T$ can be deterministic or even adversarial. However, one important drawback of Theorem 2.1 is that $\|\boldsymbol{\Delta}_T\|_{\mathrm{op}}$ must be upper bounded by $O(\lambda_{\min}/d)$, which is a strong assumption for many practical applications [28]. On the other hand, [2, 24] show that by using smart initializations the robust tensor power method is capable of tolerating $O(\lambda_{\min}/\sqrt{d})$

magnitude of noise, and [25] suggests that such noise magnitude cannot be improved if deflation (i.e., successive rank-one approximation) is to be performed.

In this paper, we show that the relaxed noise bound $O(\lambda_{\min}/\sqrt{d})$ holds even if the initialization of robust TPM is as simple as a vector uniformly sampled from the $d$-dimensional sphere (Algorithm 1). Our claim is formalized below:

**Theorem 2.2** (Improved noise tolerance analysis for robust TPM). *Assume the same notation as in Theorem 2.1. Let $\epsilon \in (0, 1/2)$ be an error tolerance parameter. There exist absolute constants $K_0, C_0, C_1, C_2, C_3 > 0$ such that if $\boldsymbol{\Delta}_T$ satisfies*

$$\|\boldsymbol{\Delta}_T(\mathbf{I}, \boldsymbol{u}_t^{(\tau)}, \boldsymbol{u}_t^{(\tau)})\|_2 \le \epsilon, \quad |\boldsymbol{\Delta}_T(\boldsymbol{v}_i, \boldsymbol{u}_t^{(\tau)}, \boldsymbol{u}_t^{(\tau)})| \le \min\{\epsilon/\sqrt{k}, C_0\lambda_{\min}/d\} \quad (4)$$

*for all $i \in [k]$, $t \in [T]$, $\tau \in [L]$ and furthermore*

$$\epsilon \le C_1 \cdot \lambda_{\min}/\sqrt{k}, \quad R = \Omega(\log(\lambda_{\max}d/\epsilon)), \quad L = \Omega(\max\{K_0, k\}\log(\max\{K_0, k\})), \quad (5)$$

*then with probability at least $0.9$, there exists a permutation $\pi : [k] \to [k]$ such that*

$$|\lambda_i - \hat{\lambda}_{\pi(i)}| \le C_2\epsilon, \quad \|\boldsymbol{v}_i - \hat{\boldsymbol{v}}_{\pi(i)}\|_2 \le C_3\epsilon/\lambda_i, \quad \forall i = 1, \cdots, k.$$

Due to space constraints, proof of Theorem 2.2 is placed in Appendix C. We next make several remarks on our results. In particular, we consider three scenarios with increasing assumptions imposed on the noise tensor $\boldsymbol{\Delta}_T$ and compare the noise conditions in Theorem 2.2 with existing results on orthogonal tensor decomposition:

1. $\boldsymbol{\Delta}_T$ *does not have any special structure*: in this case, we only have $|\boldsymbol{\Delta}_T(\boldsymbol{v}_i, \boldsymbol{u}_t, \boldsymbol{u}_t)| \le \|\boldsymbol{\Delta}_T\|_{\mathrm{op}}$ and our noise conditions reduces to the classical one: $\|\boldsymbol{\Delta}_T\|_{\mathrm{op}} = O(\lambda_{\min}/d)$.

2. $\boldsymbol{\Delta}_T$ *is "round"* in the sense that $|\boldsymbol{\Delta}_T(\boldsymbol{v}_i, \boldsymbol{u}_t, \boldsymbol{u}_t)| \le O(1/\sqrt{d}) \cdot \|\boldsymbol{\Delta}_T(\mathbf{I}, \boldsymbol{u}_t, \boldsymbol{u}_t)\|_2$: this is the typical setting when the noise $\boldsymbol{\Delta}_T$ follows Gaussian or sub-Gaussian distributions, as we explain in Sec. 3 and 4. Our noise condition in this case is $\|\boldsymbol{\Delta}_T\|_{\mathrm{op}} = O(\lambda_{\min}/\sqrt{d})$, strictly improving Theorem 2.1 on robust tensor power method with random initializations and matching the bound for more advanced SVD initialization techniques in [2].

3. $\boldsymbol{\Delta}_T$ *is weakly correlated with signal* in the sense that $\|\boldsymbol{\Delta}_T(\boldsymbol{v}_i, \mathbf{I}, \mathbf{I})\|_2 = O(\lambda_{\min}/d)$ for all $i \le k$: in this case our noise condition reduces to $\|\boldsymbol{\Delta}_T\|_{\mathrm{op}} = O(\lambda_{\min}/\sqrt{k})$, strictly improving over SVD initialization [2] in the "undercomplete" regime $k = o(d)$. Note that the whitening trick [3, 1] does not attain our bound, as we explain in Appendix B.

Finally, we remark that the $\log\log(1/\epsilon)$ quadratic convergence rate in Eq. (3) is worsened to $\log(1/\epsilon)$ linear rate in Eq. (5). We are not sure whether this is an artifact of our analysis, because similar analysis for the matrix noisy power method [12] also reveals a linear convergence rate.

**Implications** Our bounds in Theorem 2.2 results in sharper analysis of both memory-efficient and differentially private power methods which we propose in Sec. 3, 4. Using the original analysis (Theorem 2.1) for the two applications, the memory-efficient tensor power method would have sample complexity *cubic* in the dimension $d$ and for differentially private tensor decomposition the privacy level $\varepsilon$ needs to scale as $\tilde{\Omega}(\sqrt{d})$ as $d$ increases, which is particularly bad as the quality of privacy protection $e^\varepsilon$ degrades exponentially with tensor dimension $d$. On the other hand, our improved noise condition in Theorem 2.2 greatly sharpens the bounds in both applications: for memory efficient decomposition, we now require only quadratic sample complexity and for differentially private decomposition, the privacy level $\varepsilon$ has no polynomial dependence on $d$. This makes our results far more practical for high-dimensional tensor decomposition applications.

**Numerical verification of noise conditions and comparison with whitening techniques** We verify our improved noise conditions for robust tensor power method on simulation tensor data. In particular, we consider three noise models and demonstrate varied asymptotic noise magnitudes at which tensor power method succeeds. The simulation results nicely match our theoretical findings and also suggest, in an empirical way, tightness of noise bounds in Theorem 2.2. Due to space constraints, simulation results are placed in Appendix A.

We also compare our improved noise bound with those obtained by *whitening*, a popular technique that reduces tensor decomposition to matrix decomposition problems [1, 21, 28]. We show in Appendix B that, without side information the standard analysis of whitening based tensor decomposition leads to worse noise tolerance bounds than what we obtained in Theorem 2.2.

## 3 Memory-Efficient Streaming Tensor Decomposition

Tensor power method in Algorithm 1 requires significant storage to be deployed: $\Omega(d^3)$ memory is required to store a dense $d \times d \times d$ tensor, which is prohibitively large in many real-world applications as tensor dimension $d$ could be really high. We show in this section how to compute tensor decomposition in a memory efficient manner, with storage scaling *linearly* in $d$. In particular, we consider the case when tensor $\mathbf{T}$ to be decomposed is a *population moment* $\mathbb{E}_{\boldsymbol{x}\sim\mathcal{D}}[\boldsymbol{x}^{\otimes 3}]$ with respect to some unknown underlying data distribution $\mathcal{D}$, and data points $\boldsymbol{x}_1, \boldsymbol{x}_2, \cdots$ i.i.d. sampled from $\mathcal{D}$ are fed into a tensor decomposition algorithm in a streaming fashion. One classical example is topic modeling, where each $\boldsymbol{x}_i$ represents documents that come in streams and consistent estimation of topics can be achieved by decomposing variants of the population moment [1, 3].

Algorithm 2 displays memory-efficient tensor decomposition procedure on streaming data points. The main idea is to replace the power iteration step $\mathbf{T}(\mathbf{I}, \boldsymbol{u}, \boldsymbol{u})$ in Algorithm 1 with a "data association" step that exploits the empirical-moment structure of the tensor $\mathbf{T}$ to be decomposed and evaluates approximate power iterations from stochastic data samples. This procedure is highly efficient, in that both time and space complexity scale linearly with tensor dimension $d$:

**Proposition 3.1.** *Algorithm 2 runs in $O(nkdLR)$ time and $O(d(k+L))$ memory, with $O(nkR)$ sample complexity (number of data point gone through).*

In the remainder of this section we show Algorithm 2 recovers eigenvectors of the population moment $\mathbb{E}_{\boldsymbol{x}\sim\mathcal{D}}[\boldsymbol{x}^{\otimes 3}]$ with high probability and we derive corresponding sample complexity bounds. To facilitate our theoretical analysis we need several assumptions on the data distribution $\mathcal{D}$. The first natural assumption is the low-rankness of the population moment $\mathbb{E}_{\boldsymbol{x}\sim\mathcal{D}}[\boldsymbol{x}^{\otimes 3}]$ to be decomposed:

**Assumption 3.1** (Low-rank moment). *The mean tensor $\mathbf{T} = \mathbb{E}_{\boldsymbol{x}\sim\mathcal{D}}[\boldsymbol{x}^{\otimes 3}]$ admits a low-rank representation $\mathbf{T} = \sum_{i=1}^{k} \lambda_i \boldsymbol{v}_i^{\otimes 3}$ for $\lambda_1, \cdots, \lambda_k > 0$ and orthonormal $\{\boldsymbol{v}_1, \cdots, \boldsymbol{v}_k\} \subseteq \mathbb{R}^d$.*

We also place restrictions on the "noise model", which imply that the population moment $\mathbb{E}_{\boldsymbol{x}\sim\mathcal{D}}[\boldsymbol{x}^{\otimes 3}]$ can be well approximated by a reasonable number of samples with high probability. In particular, we consider sub-Gaussian noise as formulated in Definition 3.1 and Assumption 3.2:

**Definition 3.1** (Multivariate sub-Gaussian distribution, [15]). *A $D$-dimensional random variable $\boldsymbol{x}$ belongs to the sub-Gaussian distribution family $\mathcal{SG}_D(\sigma)$ with parameter $\sigma > 0$ if it has zero mean and $\mathbb{E}\left[\exp(\boldsymbol{a}^\top \boldsymbol{x})\right] \leq \exp\left\{\|\boldsymbol{a}\|_2^2 \sigma^2/2\right\}$ for all $\boldsymbol{a} \in \mathbb{R}^D$.*

**Assumption 3.2** (Sub-Gaussian noise). *There exists $\sigma > 0$ such that the mean-centered vectorized random variable $\mathrm{vec}(\boldsymbol{x}^{\otimes 3} - \mathbb{E}[\boldsymbol{x}^{\otimes 3}])$ belongs to $\mathcal{SG}_{d^3}(\sigma)$ as defined in Definition 3.1.*

We remark that Assumption 3.2 includes a wide family of distributions that are of practical importance, for example noise that have compact support. Assumption 3.2 also resembles $(B, p)$-*round noise* considered in [12] that imposes spherical symmetry constraints onto the noise distribution.

We are now ready to present the main theorem that bounds the recovery (approximation) error of eigenvalues and eigenvectors of the streaming robust tensor power method in Algorithm 2:

**Theorem 3.1** (Analysis of streaming robust tensor power method). *Let Assumptions 3.1, 3.2 hold true and suppose $\epsilon < C_1 \lambda_{\min}/\sqrt{k}$ for some sufficiently small absolute constant $C_1 > 0$. If*

$$n = \widetilde{\Omega}\left(\min\left\{\frac{\sigma^2 d}{\epsilon^2}, \frac{\sigma^2 d^2}{\lambda_{\min}^2}\right\}\right), \quad R = \Omega(\log(\lambda_{\max} d/\epsilon)), \quad L = \Omega(k \log k),$$

*then with probability at least 0.9 there exists permutation $\pi : [k] \to [k]$ such that*

$$|\lambda_i - \hat{\lambda}_{\pi(i)}| \leq C_2 \epsilon, \quad \|\boldsymbol{v}_i - \hat{\boldsymbol{v}}_{\pi(i)}\|_2 \leq C_3 \epsilon/\lambda_i, \quad \forall i = 1, \cdots, k$$

*for some universal constants $C_2, C_3 > 0$.*

Corollary 3.1 is then an immediate consequence of Theorem 3.1, which simplifies the bounds and highlights asymptotic dependencies over important model parameters $d, k$ and $\sigma$:

---

**Algorithm 2** Online robust tensor power method

---
1: **Input**: data stream $\boldsymbol{x}_1, \boldsymbol{x}_2, \cdots \in \mathbb{R}^d$, no. of components $k$, parameters $L, R, n$.
2: **for** $i = 1$ to $k$ **do**
3:     Draw $\boldsymbol{u}_0^{(1)}, \cdots, \boldsymbol{u}_0^{(L)}$ i.i.d. uniformly at random from the unit sphere $\mathcal{S}^{d-1}$.
4:     **for** $t = 0$ to $R - 1$ **do**
5:         Initialization: Set accumulators $\tilde{\boldsymbol{u}}_{t+1}^{(1)}, \cdots, \tilde{\boldsymbol{u}}_{t+1}^{(L)}$ and $\tilde{\lambda}^{(1)}, \cdots, \tilde{\lambda}^{(L)}$ to 0.
6:         Data association: Read the next $n$ data points; update $\tilde{\boldsymbol{u}}_{t+1}^{(\tau)} \leftarrow \tilde{\boldsymbol{u}}_{t+1}^{(\tau)} + \frac{1}{n}(\boldsymbol{x}_\ell^\top \boldsymbol{u}_t^{(\tau)})^2 \boldsymbol{x}_i$
        and $\tilde{\lambda}^{(\tau)} \leftarrow \tilde{\lambda}^{(\tau)} + \frac{1}{n}(\boldsymbol{x}_\ell^\top \boldsymbol{u}_t^{(\tau)})^3$ for each $\ell \in [n]$ and $\tau \in [L]$.
7:         Deflation: For each $\tau \in [L]$, update $\tilde{\boldsymbol{u}}_{t+1}^{(\tau)} \leftarrow \tilde{\boldsymbol{u}}_{t+1}^{(\tau)} - \sum_{j=1}^{i-1} \hat{\lambda}_j \xi_{j,\tau}^2 \hat{\boldsymbol{v}}_j$
        and $\tilde{\lambda}^{(\tau)} \leftarrow \tilde{\lambda}^{(\tau)} - \sum_{j=1}^{i-1} \hat{\lambda}_j \xi_{j,\tau}^3$, where $\xi_{j,\tau} = \hat{\boldsymbol{v}}_j^\top \tilde{\boldsymbol{u}}_t^{(\tau)}$.
8:         Normalization: $\boldsymbol{u}_{t+1}^{(\tau)} = \tilde{\boldsymbol{u}}_{t+1}^{(\tau)}/\|\tilde{\boldsymbol{u}}_{t+1}^{(\tau)}\|_2$, for each $\tau \in [L]$.
9:     **end for**
10:     Find $\tau^* = \operatorname{argmax}_{\tau \in [L]} \tilde{\lambda}^{(\tau)}$ and store $\hat{\lambda}_i = \tilde{\lambda}^{(\tau^*)}$, $\hat{\boldsymbol{v}}_i = \boldsymbol{u}_R^{(\tau^*)}$.
11: **end for**
12: **Output**: approximate eigenvalue and eigenvector pairs $\{\hat{\lambda}_i, \hat{\boldsymbol{v}}_i\}_{i=1}^k$ of $\hat{\mathbb{E}}_{\boldsymbol{x} \sim \mathcal{D}}[\boldsymbol{x}^{\otimes 3}]$.

---

**Corollary 3.1.** *Under Assumptions 3.1, 3.2, Algorithm 2 correctly learns $\{\lambda_i, \boldsymbol{v}_i\}_{i=1}^k$ up to $O(1/\sqrt{d})$ additive error with $\tilde{O}(\sigma^2 k d^2)$ samples and $\tilde{O}(dk)$ memory.*

Proofs of Theorem 3.1 and Corollary 3.1 are both deferred to Appendix D. Compared to streaming noisy matrix PCA considered in [12], the bound is weaker with an additional $1/k$ factor in the term involving $\epsilon$ and $1/d$ factor in the term that does not involve $\epsilon$. We conjecture this to be a fundamental difficulty of the tensor decomposition problem. On the other hand, our bounds resulting from the analysis in Sec. 2 have a $O(1/d)$ improvement compared to applying existing analysis in [1] directly.

**Remark on comparison with SGD:** Our proposed streaming tensor power method is nothing but the projected stochastic gradient descent (SGD) procedure on the objective of maximizing the tensor norm on the sphere. The optimal solution of this coincides with the objective of finding the best rank-1 approximation of the tensor. Here, we can estimate all the components of the tensor through deflation. An alternative method is to run SGD based a combined objective function to obtain all the components of the tensor simultaneously, as considered in [16, 11]. However, the analysis in [11] only works for even-order tensors and has worse dependency (at least $d^9$) on tensor dimension $d$.

## 4 Differentially private tensor decomposition

The objective of private data processing is to release data summaries such that any particular entry of the original data cannot be reliably inferred from the released results. Formally speaking, we adopt the popular $(\varepsilon, \delta)$-differential privacy criterion proposed in [9]:

**Definition 4.1** (($\varepsilon, \delta$)-differential privacy [9]). *Let $\mathcal{M}$ denote all symmetric $d$-dimensional real third order tensors and $\mathcal{O}$ be an arbitrary output set. A randomized algorithm $A : \mathcal{M} \to \mathcal{O}$ is $(\varepsilon, \delta)$-differentially private if for all neighboring tensors $\mathbf{T}, \mathbf{T}'$ and measurable set $O \subseteq \mathcal{O}$ we have*

$$\Pr\left[A(\mathbf{T}) \in O\right] \le e^\varepsilon \Pr\left[A(\mathbf{T}') \in O\right] + \delta,$$

*where $\varepsilon > 0$, $\delta \in [0, 1)$ are privacy parameters and probabilities are taken over randomness in $A$.*

Since our tensor decomposition analysis concerns symmetric tensors primarily, we adopt a "symmetric" definition of neighboring tensors in Definition 4.1, as shown below:

**Definition 4.2** (Neighboring tensors). *Two $d \times d \times d$ symmetric tensors $\mathbf{T}, \mathbf{T}'$ are neighboring tensors if there exists $i, j, k \in [d]$ such that*

$$\mathbf{T}' - \mathbf{T} = \pm\text{symmetrize}(\boldsymbol{e}_i \otimes \boldsymbol{e}_j \otimes \boldsymbol{e}_k) = \pm\left(\boldsymbol{e}_i \otimes \boldsymbol{e}_j \otimes \boldsymbol{e}_k + \boldsymbol{e}_i \otimes \boldsymbol{e}_k \otimes \boldsymbol{e}_j + \cdots + \boldsymbol{e}_k \otimes \boldsymbol{e}_j \otimes \boldsymbol{e}_i\right).$$

As noted earlier, the above notions can be similarly extended to asymmetric tensors as well as the guarantees for tensor power method on asymmetric tensors. We also remark that the difference of

---
**Algorithm 3** Differentially private robust tensor power method
---
1: **Input**: tensor $\mathbf{T}$, no. of components $k$, number of iterations $L, R$, privacy parameters $\varepsilon, \delta$.

2: **Initialization**: $\mathbf{D} = 0$, $\nu = \frac{6\sqrt{2\ln(1.25/\delta')}}{\varepsilon'}$, $\delta' = \frac{\delta}{2K}$, $\varepsilon' = \frac{\varepsilon}{\sqrt{K(4+\ln(2/\delta))}}$, $K = kL(R+1)$.

3: **for** $i = 1$ to $k$ **do**

4:     *Initialization*: Draw $\boldsymbol{u}_0^{(1)}, \cdots, \boldsymbol{u}_0^{(\tau)}$ uniformly at random from the unit sphere in $\mathbb{R}^d$.

5:     **for** $t = 0$ to $R - 1$ **do**

6:        *Power iteration*: compute $\tilde{\boldsymbol{u}}_{t+1}^{(\tau)} = (\mathbf{T} - \mathbf{D})(\mathbf{I}, \boldsymbol{u}_t^{(\tau)}, \boldsymbol{u}_t^{(\tau)})$.

7:        *Noise calibration*: release $\bar{\boldsymbol{u}}_{t+1}^{(\tau)} = \tilde{\boldsymbol{u}}_{t+1}^{(\tau)} + \nu\|\boldsymbol{u}_t^{(\tau)}\|_\infty^2 \cdot \boldsymbol{z}_t^{(\tau)}$, where $\boldsymbol{z}_t^{(\tau)} \stackrel{i.i.d.}{\sim} \mathcal{N}(\mathbf{0}, \mathbf{I}_d)$.

8:        *Normalization*: $\boldsymbol{u}_{t+1}^{(\tau)} = \bar{\boldsymbol{u}}_{t+1}^{(\tau)}/\|\bar{\boldsymbol{u}}_{t+1}^{(\tau)}\|_2$.

9:     **end for**

10:    Compute $\tilde{\lambda}^{(\tau)} = (\mathbf{T} - \mathbf{D})(\boldsymbol{u}_R^{(\tau)}, \boldsymbol{u}_R^{(\tau)}, \boldsymbol{u}_R^{(\tau)}) + \nu\|\boldsymbol{u}_R^{(\tau)}\|_\infty^3 \cdot z_\tau$ and let $\tau^* = \text{argmax}_\tau \tilde{\lambda}^{(\tau)}$.

11:    *Deflation*: $\hat{\lambda}_i = \tilde{\lambda}^{(\tau^*)}$, $\hat{\boldsymbol{v}}_i = \boldsymbol{u}_R^{(\tau^*)}$, $\mathbf{D} \leftarrow \mathbf{D} + \hat{\lambda}_i \hat{\boldsymbol{v}}_i^{\otimes 3}$.

12: **end for**

13: **Output**: eigenvalue/eigenvector pairs $\{\hat{\lambda}_i, \hat{\boldsymbol{v}}_i\}_{i=1}^k$.

---

"neighboring tensors" as defined above has Frobenious norm bounded by $O(1)$. This is necessary because an arbitrary perturbation of a tensor, even if restricted to only one entry, is capable of destroying any utility guarantee possible.

In a nutshell, Definitions 4.1, 4.2 state that an algorithm $A$ is differentially private if, conditioned on any set of possible outputs of $A$, one cannot distinguish with high probability between two "neighboring" tensors $\mathbf{T}, \mathbf{T}'$ that differ only in a single entry (up to symmetrization), thus protecting the privacy of any particular element in the original tensor $\mathbf{T}$. Here $\varepsilon, \delta$ are parameters controlling the level of privacy, with smaller $\varepsilon, \delta$ values implying stronger privacy guarantee as $\Pr[A(\mathbf{T}) \in O]$ and $\Pr[A(\mathbf{T}') \in O]$ are closer to each other.

Algorithm 3 describes the procedure of privately releasing eigenvectors of a low-rank input tensor $\mathbf{T}$. The main idea for privacy preservation is the following *noise calibration* step

$$\bar{\boldsymbol{u}}_{t+1} = \tilde{\boldsymbol{u}}_{t+1} + \nu\|\boldsymbol{u}_t\|_\infty^2 \cdot \boldsymbol{z}_t,$$

where $\boldsymbol{z}_t$ is a $d$-dimensional standard Normal random variable and $\nu\|\boldsymbol{u}_t\|_\infty^2$ is a carefully designed noise magnitude in order to achieved desired privacy level $(\varepsilon, \delta)$. One key aspect is that the noise calibration step occurs at *every* power iteration, which adds to the robustness of the algorithm and achieves sharper bounds. We discuss at the end of this section.

**Theorem 4.1** (Privacy guarantee). *Algorithm 3 satisfies $(\varepsilon, \delta)$-differential privacy.*

*Proof.* The only power iteration step of Algorithm 3 can be thought of as $K = kL(R+1)$ queries directed to a private data sanitizer which produces $f_1(\mathbf{T}; \boldsymbol{u}) = \mathbf{T}(\mathbf{I}, \boldsymbol{u}, \boldsymbol{u})$ or $f_2(\mathbf{T}; \boldsymbol{u}) = \mathbf{T}(\boldsymbol{u}, \boldsymbol{u}, \boldsymbol{u})$ each time. The $\ell_2$-sensitivity of both queries can be separately bounded as

$$\Delta_2 f_1 = \sup_{\mathbf{T}'} \|\mathbf{T}(\mathbf{I}, \boldsymbol{u}, \boldsymbol{u}) - \mathbf{T}'(\mathbf{I}, \boldsymbol{u}, \boldsymbol{u})\|_2 \leq \sup_{i,j,k} 2(|\boldsymbol{u}_i\boldsymbol{u}_j| + |\boldsymbol{u}_i\boldsymbol{u}_k| + |\boldsymbol{u}_j\boldsymbol{u}_k|) \leq 6\|\boldsymbol{u}\|_\infty^2;$$

$$\Delta_2 f_2 = \sup_{\mathbf{T}'} |\mathbf{T}(\boldsymbol{u}, \boldsymbol{u}, \boldsymbol{u}) - \mathbf{T}'(\boldsymbol{u}, \boldsymbol{u}, \boldsymbol{u})| = \sup_{i,j,k} 6|\boldsymbol{u}_i\boldsymbol{u}_j\boldsymbol{u}_k| \leq 6\|\boldsymbol{u}\|_\infty^3,$$

where $\mathbf{T}' = \mathbf{T} + \text{symmetrize}(\boldsymbol{e}_i \otimes \boldsymbol{e}_j \otimes \boldsymbol{e}_k)$ is some neighboring tensor of $\mathbf{T}$. Thus, applying the Gaussian mechanism [9] we can $(\varepsilon, \delta)$-privately release *one output* of either $f_1(\boldsymbol{u})$ or $f_2(\boldsymbol{u})$ by

$$f_\ell(\boldsymbol{u}) + \frac{\Delta_2 f_\ell \cdot \sqrt{2\ln(1.25/\delta)}}{\varepsilon} \cdot \boldsymbol{w},$$

where $\ell = 1, 2$ and $\boldsymbol{w} \sim \mathcal{N}(\mathbf{0}, \mathbf{I})$ are i.i.d. standard Normal random variables. Finally, applying *advanced composition* [9] across all $K = kL(R+1)$ private releases we complete the proof of this proposition. Note that both normalization and deflation steps do not affect the differential privacy of Algorithm 3 due to the *closeness under post-processing* property of DP. $\qquad\square$

The rest of the section is devoted to discussing the "utility" of Algorithm 3; i.e., to show that the algorithm is still capable of producing approximate eigenvectors, despite the privacy constraints. Similar to [12], we adopt the following incoherence assumptions on the eigenspace of $\mathbf{T}$:

**Assumption 4.1** (Incoherent basis). *Suppose $\mathbf{V} \in \mathbb{R}^{d \times k}$ is the stacked matrix of orthonormal component vectors $\{\boldsymbol{v}_i\}_{i=1}^k$. There exists constant $\mu_0 > 0$ such that*

$$\frac{d}{k} \max_{1 \leq i \leq d} \|\mathbf{V}^\top \boldsymbol{e}_i\|_2^2 \leq \mu_0. \tag{6}$$

Note that by definition, $\mu_0$ is always in the range of $[1, d/k]$. Intuitively, Assumption 4.1 with small constant $\mu_0$ implies a relatively "flat" distribution of element magnitudes in $\mathbf{T}$. The incoherence level $\mu_0$ plays an important role in the utility guarantee of Algorithm 3, as we show below:

**Theorem 4.2** (Guaranteed recovery of eigenvector under privacy requirements). *Suppose $\mathbf{T} = \sum_{i=1}^k \lambda_i \boldsymbol{v}_i^{\otimes 3}$ for $\lambda_1 > \lambda_2 \geq \lambda_3 \geq \cdots \geq \lambda_k > 0$ with orthonormal $\boldsymbol{v}_1, \cdots, \boldsymbol{v}_k \in \mathbb{R}^d$, and suppose Assumption 4.1 holds with $\mu_0$. Assume $\lambda_1 - \lambda_2 \geq c/\sqrt{d}$ for some sufficiently small universal constant $c > 0$. If $R = \Theta(\log(\lambda_{\max} d))$, $L = \Theta(k \log k)$ and $\varepsilon, \delta$ satisfy*

$$\varepsilon = \Omega\left(\frac{\mu_0 k^2 \log(\lambda_{\max} d/\delta)}{\lambda_{\min}}\right), \tag{7}$$

*then with probability at least 0.9 the first eigen pair $(\hat{\lambda}_1, \hat{\boldsymbol{v}}_1)$ returned by Algorithm 3 satisfies*

$$\left|\lambda_1 - \hat{\lambda}_1\right| = O(1/\sqrt{d}), \qquad \|\boldsymbol{v}_1 - \hat{\boldsymbol{v}}_1\|_2 = O(1/(\lambda_1 \sqrt{d})).$$

At a high level, Theorem 4.2 states that when the privacy parameter $\varepsilon$ is not too small (i.e., privacy requirements are not too stringent), Algorithm 3 approximately recovers the largest eigenvalue and eigenvector with high probability. Furthermore, when $\mu_0$ is a constant, the lower bound condition on the privacy parameter $\varepsilon$ does *not* depend polynomially upon tensor dimension $d$, which is a much desired property for high-dimensional data analysis. On the other hand, similar results cannot be achieved via simpler methods like input perturbation, as we discuss below:

**Comparison with input perturbation** Input perturbation is perhaps the simplest method for differentially private data analysis and has been successful in numerous scenarios, e.g. private matrix PCA [10]. In our context, this would entail appending a random Gaussian tensor $\mathbf{E}$ directly onto the input tensor $\mathbf{T}$ *before* tensor power iterations. By Gaussian mechanism, the standard deviation $\sigma$ of each element in $\mathbf{E}$ scales as $\sigma = \Omega(\varepsilon^{-1}\sqrt{\log(1/\delta)})$. On the other hand, noise analysis for tensor decomposition derived in [24, 2] and in the subsequent section of this paper requires $\sigma = O(1/d)$ or $\|\mathbf{E}\|_{\mathrm{op}} = O(1/\sqrt{d})$, which implies $\varepsilon = \tilde{\Omega}(d)$ (cf. Lemma F.9). That is, the privacy parameter $\varepsilon$ must scale *linearly* with tensor dimension $d$ to successfully recover even the first principle eigenvector, which renders the privacy guarantee of the input perturbation procedure useless for high-dimensional tensors. Thus, we require a non-trivial new approach for differentially private tensor decomposition.

Finally, we remark that a more desired utility analysis would bound the approximation error $\|\boldsymbol{v}_i - \hat{\boldsymbol{v}}_i\|_2$ for every component $\boldsymbol{v}_1, \cdots, \boldsymbol{v}_k$, and not just the top eigenvector. Unfortunately, our current analysis cannot handle deflation effectively as the deflated vector $\hat{\boldsymbol{v}}_i - \boldsymbol{v}_i$ may not be incoherent. Extension to deflated tensor decomposition remains an interesting open question.

## 5 Conclusion

We consider memory-efficient and differentially private tensor decomposition problems in this paper and derive efficient algorithms for both online and private tensor decomposition based on the popular tensor power method framework. Through an improved noise condition analysis of robust tensor power method, we obtain sharper dimension-dependent sample complexity bounds for online tensor decomposition and wider range of privacy parameters values for private tensor decomposition while still retaining utility. Simulation results verify the tightness of our noise conditions in principle.

One important direction of future research is to extend our online and/or private tensor decomposition algorithms and analysis to practical applications such as topic modeling and community detection, where tensor decomposition acts as one critical step for data analysis. An end-to-end analysis of online/private methods for these applications would be theoretically interesting and could also greatly benefit practical machine learning of important models.

**Acknowledgement** A. Anandkumar is supported in part by Microsoft Faculty Fellowship, NSF Career award CCF-1254106, ONR Award N00014- 14-1-0665, ARO YIP Award W911NF-13-1-0084 and AFOSR YIP FA9550-15-1-0221.

## Footnotes

[1] $\tilde{O}$ hides poly-logarithmic factors.

[2] One can always assume without loss of generality that $\lambda \geq 0$ by replacing $\boldsymbol{v}$ with $-\boldsymbol{v}$ instead.

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
