[Supplementary Material]



Figure 1: Failure probability against scaled noise magnitude on synthetic tensors. From top to bottom rows: random Gaussian noise, adversarial noise and noise that weakly correlate with the signals (details in main text). From left to right: scaled noise magnitudes: $\sigma$, $\sigma\sqrt{d}$, $\sigma d$ and $\sigma\ln(d)$, labeled on each plot below the $X$ axis.

## A   Simulation results

We verify our main theoretical results in Theorem 2.2 on synthetic tensors. $\mathbf{T}$ is taken to be a rank-3 tensor $\mathbf{T} = \boldsymbol{v}_1^{\otimes 3} + 0.75\boldsymbol{v}_2^{\otimes 3} + 0.5\boldsymbol{v}_3^{\otimes 3}$, where $\boldsymbol{v}_1 = (1, 0, 0, 0, \cdots, )$, $\boldsymbol{v}_2 = (0, 1, 0, 0, \cdots)$ and $\boldsymbol{v}_3 = (0, 0, 1, 0, \cdots)$. The noise tensor $\mathbf{E}$ is synthesized according to the following three regimes:

1. **Random Gaussian noise**: First generate $\mathbf{E}_{ijk} \overset{i.i.d.}{\sim} \mathcal{N}(0,1)$ and then super-symmetrize $\mathbf{E}$.

2. **Adversarial Gaussian noise**: $\mathbf{E} = \sum_{i=1}^{d} \boldsymbol{v}_2 \otimes \boldsymbol{e}_i \otimes \boldsymbol{e}_i + \boldsymbol{e}_i \otimes \boldsymbol{v}_2 \otimes \boldsymbol{e}_i + \boldsymbol{e}_i \otimes \boldsymbol{e}_i \otimes \boldsymbol{v}_2$, where $\boldsymbol{e}_i = (0, \cdots, 0, 1, 0, \cdots, 0)$ has all zero entries except for the $i$th one.

3. **Weakly correlated noise**: Let $\{\boldsymbol{v}_4, \cdots, \boldsymbol{v}_d\}$ be an orthonormal basis of the orthogonal complement of $\mathrm{span}\{\boldsymbol{v}_1, \boldsymbol{v}_2, \boldsymbol{v}_3\}$. Set $\mathbf{E} = \sum_{i=4}^{d} \boldsymbol{v}_i \otimes \boldsymbol{v}_i \otimes \boldsymbol{v}_i$.

In Fig. 1 we plot the "failure probability" (measured via 20 independent trials per setting) of the robust tensor power method with random initialization against controlled noise magnitude $\|\mathbf{E}\|_{\mathrm{op}} = \sigma$. A trial is "successful" if for all $i \in \{1, 2, 3\}$ the recovered eigenvector $\hat{\boldsymbol{v}}_i$ satisfies $\hat{\boldsymbol{v}}_i^\top \boldsymbol{v}_i \geq 1/4$. To control $\|\mathbf{E}\|_{\mathrm{op}}$, we first compute the operator norm of the generated raw noise tensor by invoking the `eig_sshopm` routine in Matlab tensor toolbox [5] (algorithm based on [20]) and then re-scale the entries. By inspecting the noise levels at which phase transition of failure probabilities occurs for different tensor dimensions $d$, ranging from 25 to 200. It is quite clear from Fig. 1 that the phase transitions occur at $\sigma = O(1/\sqrt{d})$ for random Gaussian noise, $\sigma = O(1/d)$ for adversarial noise and $\sigma = O(1/\log d)$ for weakly correlated noises, which matches our theoretical findings in Sec. 2 up to logarithmic terms. Our simulation results and explicit construction of an "adversarial" noise matrix also suggests that our analysis for robust tensor power method with random initializations under random Gaussian noise and existing analysis for worst-case noise in [1] are tight.

## B   Comparison with whitening and matrix SVD decompositions

Another popular thread of tensor decomposition techniques involve whitening and reducing the problem to a matrix SVD decomposition, which is very effective at reducing the dimensionality of

the problem in the $k = o(d)$ undercomplete settings [1, 21, 28]. We show in this section that *without additional side information*, a standard application and analysis of tensor decomposition of whitening and matrix SVD techniques leads to *worse* error bounds than we established in Theorem 2.2.

When *only* the 3rd-order tensor $\mathbf{T}$ is available, one common whitening approach is to randomly "marginalized out" one view of $\widetilde{\mathbf{T}}$:

$$\mathbf{M}(\boldsymbol{\theta}) := \widetilde{\mathbf{T}}(\mathbf{I}, \mathbf{I}, \boldsymbol{\theta}), \qquad \boldsymbol{\theta} \text{ randomly drawn on the unit } d\text{-dimensional sphere;}$$

and then evaluate top-$k$ eigen-decomposition of $\mathbf{M}(\boldsymbol{\theta})$. Let $\mathcal{W} = \mathrm{span}(\boldsymbol{v}_1, \cdots, \boldsymbol{v}_k)$ be the span of the true components of $\mathbf{T}$ and $\hat{\mathcal{W}}$ be the top-$k$ eigenspace of matrix $\mathbf{M}(\boldsymbol{\theta})$ obtained by collapsing one view of $\widetilde{\mathbf{T}}$. We then have the following proposition that bounds the perturbation between $\mathcal{W}$ and $\hat{\mathcal{W}}$:

**Proposition B.1.** *Suppose* $\widetilde{\mathbf{T}} = \mathbf{T} + \boldsymbol{\Delta}_T$ *as in Theorems 2.1, 2.2 and let* $\boldsymbol{\Pi}_{\mathcal{W}}, \boldsymbol{\Pi}_{\hat{\mathcal{W}}}$ *be the projection operators of* $\mathcal{W}$ *and* $\hat{\mathcal{W}}$, *respectively. Then with probability at least 0.9 over the random draw of* $\boldsymbol{\theta}$,

$$\left\| \boldsymbol{\Pi}_{\mathcal{W}} - \boldsymbol{\Pi}_{\hat{\mathcal{W}}} \right\|_2 \leq \widetilde{O}\left( \frac{\sqrt{d}\|\boldsymbol{\Delta}_T\|_{\mathrm{op}}}{\lambda_{\min}} \right), \quad \text{if} \quad \|\boldsymbol{\Delta}_T\|_{\mathrm{op}} = \widetilde{O}\left( \frac{\lambda_{\min}}{\sqrt{d}} \right),$$

*Proof.* First, we decompose $\mathbf{M}(\boldsymbol{\theta})$ into two terms:

$$\mathbf{M}(\boldsymbol{\theta}) = \sum_{i=1}^{k} \lambda_i(\boldsymbol{v}_i^\top \boldsymbol{\theta}) \cdot \boldsymbol{v}_i \boldsymbol{v}_i^\top + \boldsymbol{\Delta}_T(\mathbf{I}, \mathbf{I}, \boldsymbol{\theta}).$$

Define $\bar{\lambda}_i = \lambda_i(\boldsymbol{v}_i^\top \boldsymbol{\theta})$ and $\bar{\mathbf{E}} = \boldsymbol{\Delta}_T(\mathbf{I}, \mathbf{I}, \boldsymbol{\theta})$. We then have that

$$\mathbf{M}(\boldsymbol{\theta}) = \mathbf{M}_0 + \bar{\mathbf{E}},$$

where $\mathbf{M}_0$ is a $d \times d$ rank-$k$ matrix with eigenvalues $(\bar{\lambda}_1, \cdots, \bar{\lambda}_k)$ and eigenvectors $(\boldsymbol{v}_1, \cdots, \boldsymbol{v}_k)$, and $\bar{\mathbf{E}}$ satisfies $\|\bar{\mathbf{E}}\|_2 \leq \|\boldsymbol{\Delta}_T\|_{\mathrm{op}}$. Since $\boldsymbol{\theta}$ is uniformly sampled from the $d$-dimensional unit sphere, by standard concentration arguments we have that $|\boldsymbol{v}_j^\top \boldsymbol{\theta}| = \widetilde{\Omega}(1/\sqrt{d})$ with overwhelming probability for all $j = 1, \cdots, k$ and hence

$$\sigma_k(\mathbf{M}_0) = \widetilde{\Omega}(\lambda_{\min}/\sqrt{d}),$$

where $\sigma_k(\cdot)$ denotes the $k$th largest singular value of a matrix. Applying Weyl's theorem (Lemma F.7) we have that

$$\sigma_k(\mathbf{M}(\boldsymbol{\theta})) \geq \sigma_k(\mathbf{M}_0) - \|\bar{\mathbf{E}}\|_2 = \widetilde{\Omega}(\lambda_{\min}/\sqrt{d}),$$

where the last inequality is due to the condition imposed on noise magnitude $\|\boldsymbol{\Delta}_T\|_{\mathrm{op}}$ and the fact that $\|\bar{\mathbf{E}}\|_2 \leq \|\boldsymbol{\Delta}_T\|_{\mathrm{op}}$. Applying Wedin's theorem (Lemma F.8) with $\alpha = 0$ and $\delta = \sigma_k(\mathbf{M}(\boldsymbol{\theta})) = \widetilde{\Omega}(\lambda_{\min}/\sqrt{d})$ we arrive at

$$\left\| \boldsymbol{\Pi}_{\mathcal{W}} - \boldsymbol{\Pi}_{\hat{\mathcal{W}}} \right\|_2 \leq \frac{\|\bar{\mathbf{E}}\|_2}{\sigma_k(\mathbf{M}(\boldsymbol{\theta}))} \leq \widetilde{O}\left( \frac{\sqrt{d}\|\boldsymbol{\Delta}_T\|_{\mathrm{op}}}{\lambda_{\min}} \right).$$

$\square$

This simple result shows that the whitening trick does not trivially lead to matching noise conditions in Theorem 2.2 under $k = o(d)$ settings.

## C   Proof of Theorem 2.2

### C.1   Proof sketch of Theorem 2.2

In this section we sketch the proof of Theorem 2.2. Our proof is mostly built upon the analysis in [1] for robust tensor power method. However, we borrow new ideas from [12] to substantially revise the per-iteration analysis (Lemma C.2), which subsequently results in desired relaxation of noise conditions. Some results and arguments in [1], especially those involved with absolute constants, are simplified for accessibility purposes.

We start with Lemma C.1 that analyzes random initializations against eigenvectors.

**Lemma C.1.** *Fix $j^* \in \{1, \cdots, k\}$ and $\eta \in (0, 1/2)$. Suppose $L$ satisfies $L = \Omega(k/\eta)$. Then with probability at least $1 - \eta$ there exists a initialization $\boldsymbol{u}_0$ such that*

$$\max_{1 \leq j \leq k, j \neq j^*} |\boldsymbol{v}_j^\top \boldsymbol{u}_0| \leq 0.5 |\boldsymbol{v}_{j^*}^\top \boldsymbol{u}_0| \quad and \quad |\boldsymbol{v}_{j^*}^\top \boldsymbol{u}_0| \geq 1/\sqrt{d}. \tag{8}$$

Roughly speaking, Lemma C.1 shows that with $L = \Omega(d \log d)$ initializations the initial vector $\boldsymbol{u}_0$ will slightly bias towards one of the directions $j^*$ with overwhelming probability. The lemma is a slight generalization of Lemma B.1 in [1] to the $k \leq d$ case and their proofs are similar. For completeness purposes we include its proof in Appendix C.2. Applying a standard boosting argument we have the following corollary, which guarantees exponentially decaying failure probabilities:

**Corollary C.1.** *For any $\tilde{\eta} \in (0, 1/2)$, with $L = \Omega(k \log(1/\tilde{\eta}))$ initializations Eq. (8) holds for at least one initialization with probability at least $1 - \tilde{\eta}$.*

The following lemma is the key lemma that characterizes the recovery of *single* eigenvectors of the robust tensor power method.

**Lemma C.2.** *Suppose $\lambda_1 \geq \lambda_2 \geq \cdots \geq \lambda_k \geq 0$ and assume without loss of generality that the conditions in Lemma C.1 hold with respect to $j^* = 1$. Assume in addition that*

$$\|\boldsymbol{\Delta}_T(\mathbf{I}, \boldsymbol{u}_t, \boldsymbol{u}_t)\|_2 \leq \min\left\{\tilde{\epsilon}_t, \frac{\lambda_1}{40\sqrt{d}}\right\}, \quad |\boldsymbol{\Delta}_T(\boldsymbol{v}_j, \boldsymbol{u}_t, \boldsymbol{u}_t)| \leq \min\left\{\frac{\tilde{\epsilon}_t}{\sqrt{k}}, \frac{\lambda_1}{8d}\right\}, \quad \tilde{\epsilon}_t \leq \frac{\lambda_1}{200}$$

*for all $t \in [T]$ and $j$ such that $\lambda_j > 0$. We then have that* [3]

$$\max_{j \neq 1} \lambda_j |\boldsymbol{v}_j^\top \boldsymbol{u}_t| \leq 0.5\lambda_1 |\boldsymbol{v}_1^\top \boldsymbol{u}_t|, \qquad \tan \theta(\boldsymbol{v}_1, \boldsymbol{u}_t) \leq 0.8 \tan \theta(\boldsymbol{v}_1, \boldsymbol{u}_{t-1}) + 8\tilde{\epsilon}_t/\lambda_1. \tag{9}$$

*In addition, if $\theta(\boldsymbol{v}_1, \boldsymbol{u}_t) \leq \pi/3$ we have further that*

$$\frac{|\boldsymbol{v}_j^\top \boldsymbol{u}_{t+1}|}{|\boldsymbol{v}_1^\top \boldsymbol{u}_{t+1}|} \leq 0.8 \frac{|\boldsymbol{v}_j^\top \boldsymbol{u}_t|}{|\boldsymbol{v}_1^\top \boldsymbol{u}_t|} + \frac{8\tilde{\epsilon}_t}{\lambda_1 \sqrt{k}}, \qquad \forall j > 1 \text{ and } \lambda_j > 0. \tag{10}$$

Compared to existing analysis in (Propositions B.1, B.2, Lemmas B.2, B.3, B.4 in [1]), our proof in Appendix C.2 analyzes the two-phase behavior of robust tensor power method in a unified framework and is thus much cleaner. Furthermore, we borrow ideas from [12] to prove shrinkage of the tangent angle between $\boldsymbol{v}_1$ and $\boldsymbol{u}_t$, which subsequently leads to relaxed noise conditions. We also prove additional bounds regarding $|\boldsymbol{v}_j^\top \boldsymbol{u}_t|$ for $j > 1$ to facilitate later deflation analysis. This result is used for relaxing noise conditions only and is hence not proved in previous work [1].

Finally, we present the following lemma that analyzes the deflation step in the robust noisy power method, in which both "element-wise" and "full-vector" conditions on the deflated tensor are proved.

**Lemma C.3.** *Let $\{\hat{\lambda}_i, \hat{\boldsymbol{v}}_i\}_{i=1}^k$ be eigenvalue and (orthonormal) eigenvector pairs that approximates $\{\lambda_i, \boldsymbol{v}_i\}_{i=1}^k$ with $\lambda_1 \geq \cdots \geq \lambda_k > 0$ such that for all $i \in [k]$,*

$$|\hat{\lambda}_i - \lambda_i| \leq C\epsilon, \quad \tan \theta(\boldsymbol{v}_i, \boldsymbol{v}_i) \leq \min\{\sqrt{2}, C\epsilon/\lambda_i\} \quad |\hat{\boldsymbol{v}}_i^\top \boldsymbol{v}_j| \leq C\epsilon/(\lambda_i \sqrt{k}), \;\; \forall j > i \tag{11}$$

*for some absolute constant $C > 0$ and error tolerance parameter $\epsilon > 0$. Denote $\mathbf{E}_i = \hat{\lambda}_i \hat{\boldsymbol{v}}_i^{\otimes 3} - \lambda_i \boldsymbol{v}_i^{\otimes 3}$ as the $i$th reconstruction error tensor. Let $\delta \in (0, 1)$ be an arbitrary small constant. There exist universal constants $C > 0$ such that if $\epsilon \leq C' \lambda_{\min}/\sqrt{k}$ then the following holds for all $t \in [k]$ and $\|\boldsymbol{u}\|_2 = 1$:*

$$\left\|\sum_{i=1}^t \mathbf{E}_i(\mathbf{I}, \boldsymbol{u}, \boldsymbol{u})\right\|_2 \leq \kappa_t(\boldsymbol{u})\epsilon \quad and \quad \left|\sum_{i=1}^t \mathbf{E}_i(\boldsymbol{v}_j, \boldsymbol{u}, \boldsymbol{u})\right| \leq \kappa_t(\boldsymbol{u})^2 \frac{\epsilon}{\sqrt{k}}, \;\; \forall j > t, \tag{12}$$

*where $\kappa_t(\boldsymbol{u}) = \sqrt{\delta + C'' \sum_{i=1}^t |\boldsymbol{v}_i^\top \boldsymbol{u}|^2}$ and $C'' > 0$ is a universal constant.*

We are now ready to prove the main theorem.

*Proof of Theorem 2.2.* We use induction to prove the theorem. For $i = 1$ all conditions in Lemma C.2 are satisfied with $\tilde{\epsilon}_t = 2\epsilon$ when $\epsilon \leq C_1\lambda_{\min}/\sqrt{k}$ for some sufficiently small constant $C_1 > 0$. Lemma C.2 then asserts that, with $L = \Omega(d \log d)$ initializations and $R = \Omega(\log(\lambda_1 k/\epsilon))$ iterations, $\|\hat{\boldsymbol{v}}_1 - \boldsymbol{v}_1\|_2 \leq \tan\theta(\hat{\boldsymbol{v}}_1, \boldsymbol{v}_1) \leq C_2\epsilon/\lambda_1$ for some universal constant $C_2 > 0$. Furthermore,

$$|\hat{\lambda}_1 - \lambda_1| = \left|\widetilde{\mathbf{T}}(\hat{\boldsymbol{v}}_1, \hat{\boldsymbol{v}}_1, \hat{\boldsymbol{v}}_1) - \lambda_1\right| \leq \left|\boldsymbol{\Delta}_T(\hat{\boldsymbol{v}}_1, \hat{\boldsymbol{v}}_1, \hat{\boldsymbol{v}}_1)\right| + \left|\mathbf{T}(\hat{\boldsymbol{v}}_1, \hat{\boldsymbol{v}}_1, \hat{\boldsymbol{v}}_1) - \lambda_1\right|$$

$$\leq O\left(\frac{\epsilon}{\sqrt{k}}\right) + \left|\lambda_1|\boldsymbol{v}_1^\top\hat{\boldsymbol{v}}_1|^3 - \lambda_1 + \sum_{j>1}\lambda_j|\boldsymbol{v}_j^\top\hat{\boldsymbol{v}}_1|^3\right|$$

$$\leq O\left(\frac{\epsilon}{\sqrt{k}}\right) + \left|\lambda_1\left[1 + O\left(\frac{\epsilon}{\lambda_1}\right)\right] - \lambda_1 + \sum_{j>1}\lambda_j O\left(\frac{\epsilon^3}{\lambda_j^3 k^{1.5}}\right)\right|$$

$$\leq O(\epsilon), \quad \text{if } \epsilon \leq C_1\lambda_{\min}/\sqrt{k} \text{ for some sufficiently small constant } C_1.$$

We next prove the theorem for the case of $i + 1$ assuming by induction that the theorem holds for all $\{\lambda_j, \boldsymbol{v}_j\}_{j=1}^i$. In this case, the "new" noise tensor $\widetilde{\boldsymbol{\Delta}}_T$ comes from both the original noise and also noise introduced by deflations; that is, $\widetilde{\boldsymbol{\Delta}}_T = \widetilde{\mathbf{T}} + \sum_{j=1}^i \mathbf{E}_i$. Invoking Lemma C.3 we have that $\widetilde{\boldsymbol{\Delta}}_T$ satisfies conditions in Lemma C.2 with

$$\tilde{\epsilon}_t = \epsilon\left(1 + \max\{\kappa_i(\boldsymbol{u}_t), \kappa_i(\boldsymbol{u}_t)^2\}\right),$$

where $\kappa_i(\boldsymbol{u}) = \sqrt{\delta + C''\sum_{j=1}^i |\boldsymbol{u}^\top\boldsymbol{v}_j|^2}$ as defined in Lemma C.3, provided that $\epsilon \leq C_1\lambda_{\min}/\sqrt{k}$ for some sufficiently small constant $C_1$. Furthermore, note that for arbitrary $\delta \in (0,1)$, we can again pick $C_1' > 0$ to be a sufficiently small constant (possibly depending on $\delta$) such that $\epsilon \leq C_1'\lambda_{\min}/\sqrt{k}$ would imply $\tilde{\epsilon}_t \leq \min\{\lambda_1/200, 0.01\lambda_{\min}\sqrt{\delta/(C''k)}\}$. Subsequently, by Eq. (9) we know that after $\Omega(\log(\lambda_{\max}k/\epsilon))$ iterations we have that $\tan\theta(\boldsymbol{u}_t, \boldsymbol{v}_{i+1}) \leq 0.1\sqrt{\delta/(C''k)}$ and hence for any $j \leq i$, $|\boldsymbol{u}_t^\top\boldsymbol{v}_j| = \cos\theta(\boldsymbol{u}_t, \boldsymbol{v}_j) = \sin\theta(\boldsymbol{u}_t, \boldsymbol{v}_{i+1}) \leq \tan\theta(\boldsymbol{u}_t, \boldsymbol{v}_{i+1}) \leq 0.1\sqrt{\delta/(C''k)}$. Consequently, $C''\sum_{j=1}^i |\boldsymbol{u}_t^\top\boldsymbol{v}_j|^2 \leq 0.01\delta$ and therefore $\kappa_i(\boldsymbol{u}_t) \leq \sqrt{1.01\delta} \leq 1$. We then have that $\tilde{\epsilon}_t \leq 2\epsilon$ and hence the resuling bounds on $|\hat{\lambda}_{i+1} - \lambda_{i+1}|$ and $\tan\theta(\boldsymbol{u}_t, \boldsymbol{v}_{i+1})$ hold with the same constant $C$ as all previous iterations before $i$. Finally, applying Lemma C.1 and taking a union bound over all $k$ iterations we complete the proof. $\qquad\square$

## C.2 Proof of technical lemmas

*Proof of Lemma C.1.* Let $\tilde{\boldsymbol{u}}_0^{(\tau)} \overset{i.i.d.}{\sim} \mathcal{N}_d(0, \mathbf{I}_{d\times d})$ for $\tau \in [L]$ and define $Z_{j,\tau} = \boldsymbol{v}_j^\top\tilde{\boldsymbol{u}}_0^{(\tau)}$ for $j \in [d]$ and $\tau \in [L]$. Without loss of generality, assume $j^* = 1$. Consider the following sets of events:

$$\mathcal{E}_1 := \left\{Z : \max_{\tau\in[L]}|Z_{1,\tau}| \geq 0.5\sqrt{\ln L} - \sqrt{2\ln(6/\eta)}\right\}, \tag{13}$$

$$\mathcal{E}_{2,\tau} := \left\{Z_{\cdot,\tau} : \max_{1<j\leq k}|Z_{j,\tau}| \leq \sqrt{2\ln k} + \sqrt{2\ln(3/\eta)}\right\}, \tag{14}$$

$$\mathcal{E}_{3,\tau} := \left\{Z_{\cdot,\tau} : \sum_{j=k+1}^d |Z_{j,\tau}|^2 \leq 3\ln(3/\eta)\cdot d + 2\ln(3/\eta)\right\}. \tag{15}$$

Suppose $\mathcal{E}_1$ holds with $\tau^* = \operatorname{argmax}_\tau|Z_{1,\tau}|$ and suppose in addition that $\mathcal{E}_{2,\tau^*}$ and $\mathcal{E}_{3,\tau^*}$ hold. To derive Eq. (8) we need to show the following inequalities:

$$0.5\sqrt{\ln L} - \sqrt{2\ln(6/\eta)} \geq 0.5\left(\sqrt{2\ln k} + \sqrt{2\ln(3/\eta)}\right);$$

$$\frac{(0.6\sqrt{\ln L} - \sqrt{2\ln(6/\eta)})^2}{k\cdot(0.6\sqrt{\ln L} - \sqrt{2\ln(6/\eta)})^2 + 3\ln(3/\eta)d + 2\ln(3/\eta)} \geq \frac{1}{d}.$$

It can be easily verified that $L = \Omega(k/\eta)$ satisfies the above inequalities and hence imply Eq. (8) under $\mathcal{E}_1 \cap \mathcal{E}_{2,\tau^*} \cap \mathcal{E}_{3,\tau^*}$.

The rest of the proof is to lower bound the probabilities of events $\mathcal{E}_1, \mathcal{E}_{2,\tau^*}$ and $\mathcal{E}_{3,\tau^*}$. We first consider $\mathcal{E}_1$. Because $Z_{1,1}, \cdots, Z_{1,L} \overset{i.i.d.}{\sim} \mathcal{N}(0,1)$ and $f(Z_{1,1}, \cdots, Z_{1,L}) = \max_\tau |Z_{1,\tau}|$ is a 1-Lipschitz function, applying Lemma F.1 we have that

$$\Pr\left[\max_\tau |Z_{1,\tau}| < \mu - t\right] \leq 2e^{-t^2/2}, \tag{16}$$

where $\mu = \mathbb{E}[\max_\tau |Z_{1,\tau}|]$. By Lemma F.2, $\mu \geq \mathbb{E}[\max_\tau Z_{1,\tau}] \geq \sqrt{\ln L}/\sqrt{\pi \ln 2} \geq 0.5\sqrt{\ln L}$. Setting $t = \sqrt{2\ln(6/\eta)}$ in Eq. (16) we have that $\Pr[\mathcal{E}_1] \geq 1 - \eta/3$.

Next, suppose $\mathcal{E}_1$ holds with $\tau^* = \operatorname{argmax}_\tau |Z_{1,\tau}|$. Note that $\mathcal{E}_{2,\tau^*}$ and $\mathcal{E}_{3,\tau^*}$ are independent regardless of the choice of $\tau^*$, because $Z_{1,\tau^*}, \cdots, Z_{d,\tau^*}$ are independent Gaussian random variables. We can then lower bound the probabilities of $\mathcal{E}_{2,\tau^*}$ and $\mathcal{E}_{3,\tau^*}$ separately. We consider $\mathcal{E}_{2,\tau^*}$ first. Because $Z_{2,\tau^*}, \cdots, Z_{k,\tau^*}$ are i.i.d. standard Normal random variables, applying Lemma F.3 we obtain

$$\Pr\left[\max_{2 \leq j \leq k} |Z_{j,\tau^*}| > \sqrt{2\ln k} + \sqrt{2t}\right] \leq e^{-t}. \tag{17}$$

Putting $t = \ln(3/\eta)$ in Eq. (17) we have that $\Pr[\mathcal{E}_{2,\tau^*}|\mathcal{E}_1] \geq 1 - \eta/3$. For $\mathcal{E}_{3,\tau^*}$, it is obvious by definition that $\sum_{j=k+1}^{d} |Z_{j,\tau^*}|^2$ is a $\chi^2_{d-k}$-distributed random variable and is independent of $\mathcal{E}_1$ and $\mathcal{E}_{2,\tau^*}$. Applying Lemma F.4 the following holds:

$$\Pr\left[\sum_{j=k+1}^{d} |Z_{j,\tau^*}|^2 > d + 2\sqrt{dt} + 2t\right] \leq e^{-t}. \tag{18}$$

Putting $t = \ln(3/\eta)$ in Eq. (18) and noting that $\sqrt{d} \leq d$, $t \geq 1$, we conclude that $\Pr[\mathcal{E}_{3,\tau^*}|\mathcal{E}_1] \geq 1 - \eta/3$. Finally, applying union bound we have that $\Pr[\mathcal{E}_1 \cap \mathcal{E}_{2,\tau^*} \cap \mathcal{E}_{3,\tau^*}] \geq 1 - \eta$. $\qquad \square$

*Proof of Lemma C.2.* First, as a consequence of Corollary C.1, we know that $|\boldsymbol{v}_1^\top \boldsymbol{u}_0| \geq 1/\sqrt{d}$. The conditions in Lemma C.2 then imply $|\boldsymbol{\Delta}_T(\boldsymbol{v}_j, \boldsymbol{u}_t, \boldsymbol{u}_t)| \leq \lambda_1 |\boldsymbol{v}_1^\top \boldsymbol{u}_0|^2/8$. We now use induction to prove Eq. (9). When $t = 0$ Eq. (9) trivially holds due to Lemma C.1 and the condition that $j^* = 1$ corresponds to the largest eigenvalue $\lambda_1$. The objective is then to prove Eq. (9) for the case of $t+1$, assuming it holds for all iterations up to $t$.

We first consider the second part of Eq. (9) concerning $\tan \theta(\boldsymbol{v}_1, \boldsymbol{u}_t)$. Let $\mathbf{V} \in \mathbb{R}^{d \times (k-1)}$ be an orthonormal basis of the complement subspace $\mathcal{V}_\perp = \operatorname{span}\{\boldsymbol{v}_2, \cdots, \boldsymbol{v}_k\}$. Further let $\boldsymbol{\varepsilon}_t = \boldsymbol{\Delta}_T(\mathbf{I}, \boldsymbol{u}_t, \boldsymbol{u}_t)$. We then have that

$$\begin{aligned}
\tan \theta(\boldsymbol{v}_1, \boldsymbol{u}_{t+1}) &= \tan \theta(\boldsymbol{v}_1, \mathbf{T}(\mathbf{I}, \boldsymbol{u}_t, \boldsymbol{u}_t) + \boldsymbol{\varepsilon}_t) \\
&= \frac{\|\mathbf{V}^\top [\mathbf{T}(\mathbf{I}, \boldsymbol{u}_t, \boldsymbol{u}_t) + \boldsymbol{\varepsilon}_t]\|_2}{|\boldsymbol{v}_1^\top [\mathbf{T}(\mathbf{I}, \boldsymbol{u}_t, \boldsymbol{u}_t) + \boldsymbol{\varepsilon}_t]|} \\
&\leq \frac{\|\mathbf{V}^\top \mathbf{T}(\mathbf{I}, \boldsymbol{u}_t, \boldsymbol{u}_t)\|_2 + \|\mathbf{V}^\top \boldsymbol{\varepsilon}_t\|_2}{|\boldsymbol{v}_1^\top \mathbf{T}(\mathbf{I}, \boldsymbol{u}_t, \boldsymbol{u}_t)| - |\boldsymbol{v}_1^\top \boldsymbol{\varepsilon}_t|}.
\end{aligned}$$

In addition, note that

$$\|\mathbf{V}^\top \mathbf{T}(\mathbf{I}, \boldsymbol{u}_t, \boldsymbol{u}_t)\|_2 = \sqrt{\sum_{j=2}^{k} \lambda_j^2 |\boldsymbol{v}_j^\top \boldsymbol{u}_t|^4} \leq \max_{j \neq 1} \lambda_j |\boldsymbol{v}_j^\top \boldsymbol{u}_t| \cdot \sqrt{\sum_{j=2}^{k} |\boldsymbol{v}_j^\top \boldsymbol{u}_t|^2},$$

where the first equality is due to the orthogonality of $\{\boldsymbol{v}_2, \cdots, \boldsymbol{v}_k\}$ and in the last inequality we apply H'older's inequality. Because $\sqrt{\sum_{j=2}^{k} |\boldsymbol{v}_j^\top \boldsymbol{u}_t|^2} = \|\mathbf{V}^\top \boldsymbol{u}_t\|_2$, we have that

$$\begin{aligned}
\tan \theta(\boldsymbol{v}_1, \boldsymbol{u}_{t+1}) &\leq \frac{\|\mathbf{V}^\top \boldsymbol{u}_t\|_2 \cdot \max_{j \neq 1} \lambda_j |\boldsymbol{v}_j^\top \boldsymbol{u}_t| + \|\mathbf{V}^\top \boldsymbol{\varepsilon}_t\|_2}{|\boldsymbol{v}_1^\top \boldsymbol{u}_t| \cdot \lambda_1 |\boldsymbol{v}_1^\top \boldsymbol{u}_t| - |\boldsymbol{v}_1^\top \boldsymbol{\varepsilon}_t|} \\
&= \tan \theta(\boldsymbol{v}_1, \boldsymbol{u}_t) \left[ \frac{\max_{j \neq 1} \lambda_j |\boldsymbol{v}_j^\top \boldsymbol{u}_t| + \|\mathbf{V}^\top \boldsymbol{\varepsilon}_t\|_2/\|\mathbf{V}^\top \boldsymbol{u}_t\|_2}{\lambda_1 |\boldsymbol{v}_1^\top \boldsymbol{u}_t| - |\boldsymbol{v}_1^\top \boldsymbol{\varepsilon}_t|/|\boldsymbol{v}_1^\top \boldsymbol{u}_t|} \right]
\end{aligned}$$

$$\leq \tan\theta(\boldsymbol{v}_1,\boldsymbol{u}_t)\left[\frac{0.5\lambda_1|\boldsymbol{v}_1^\top\boldsymbol{u}_t| + \|\mathbf{V}^\top\varepsilon_t\|_2/\|\mathbf{V}^\top\boldsymbol{u}_t\|_2}{\lambda_1|\boldsymbol{v}_1^\top\boldsymbol{u}_t| - |\boldsymbol{v}_1^\top\varepsilon_t|/|\boldsymbol{v}_1^\top\boldsymbol{u}_t|}\right] \tag{19}$$

$$= \tan\theta(\boldsymbol{v}_1,\boldsymbol{u}_t)\underbrace{\left[\frac{1}{2}\frac{1}{1-|\boldsymbol{v}_1^\top\varepsilon_t|/(\lambda_1|\boldsymbol{v}_1^\top\boldsymbol{u}_t|^2)}\right]}_{\alpha} + \underbrace{\frac{1}{1-|\boldsymbol{v}_1^\top\varepsilon_t|/(\lambda_1|\boldsymbol{v}_1^\top\boldsymbol{u}_t|^2)}}_{2\alpha}\cdot\underbrace{\frac{\|\mathbf{V}^\top\varepsilon_t\|_2}{\lambda_1|\boldsymbol{v}_1^\top\boldsymbol{u}_t|^2}}_{\beta}\cdot$$

Here in Line 19 we apply the induction hypothesis that $\max_{j\neq 1}\lambda_j|\boldsymbol{v}_j^\top\boldsymbol{u}_t| \leq 0.5\lambda_1|\boldsymbol{v}_1^\top\boldsymbol{u}_t|$. Before proceeding the analysis we first show that $|\boldsymbol{v}_1^\top\boldsymbol{u}_0| \leq |\boldsymbol{v}_1^\top\boldsymbol{u}_t|$. Applying the induction hypothesis, we have that

$$\tan\theta(\boldsymbol{v}_1,\boldsymbol{u}_t) \leq 0.8^t\tan\theta(\boldsymbol{v}_1,\boldsymbol{u}_0) + 40\tilde{\epsilon}_t/\lambda_1 \leq 0.8\tan\theta(\boldsymbol{v}_1,\boldsymbol{u}_0) + 40\tilde{\epsilon}_t/\lambda_1 \leq \tan\theta(\boldsymbol{v}_1,\boldsymbol{u}_0),$$

where the last inequality is due to $\tilde{\epsilon}_t \leq \lambda_1/200$. Subsequently, $\theta(\boldsymbol{v}_1,\boldsymbol{u}_t) \leq \theta(\boldsymbol{v}_1,\boldsymbol{u}_0)$ and hence $|\boldsymbol{v}_1^\top\boldsymbol{u}_t| = \cos\theta(\boldsymbol{v}_1,\boldsymbol{u}_t) \geq \cos\theta(\boldsymbol{v}_1,\boldsymbol{u}_0) = |\boldsymbol{v}_1^\top\boldsymbol{u}_0|$. Now applying $|\boldsymbol{v}_1^\top\varepsilon_t| \leq |\boldsymbol{v}_1^\top\boldsymbol{u}_0|^2/4$ we obtain

$$|\boldsymbol{v}_1^\top\varepsilon_t| \leq \frac{\lambda_1|\boldsymbol{v}_1^\top\boldsymbol{u}_0|^2}{4} \leq \frac{\lambda_1|\boldsymbol{v}_1^\top\boldsymbol{u}_t|^2}{4} \implies \frac{1}{1-|\boldsymbol{v}_1^\top\varepsilon_t|/(\lambda_1|\boldsymbol{v}_1^\top\boldsymbol{u}_t|^2)} \leq \frac{3}{2} \implies \alpha \leq \frac{3}{4}. \tag{20}$$

Next we bound $\beta$ by considering two cases. In the first case of $|\boldsymbol{v}_1^\top\boldsymbol{u}_t| \leq 0.5$, we have that

$$\beta = \tan\theta(\boldsymbol{v}_1,\boldsymbol{u}_t)\cdot\frac{\|\mathbf{V}^\top\varepsilon_t\|_2}{\lambda_1|\boldsymbol{v}_1^\top\boldsymbol{u}_t|\sqrt{1-|\boldsymbol{v}_1^\top\boldsymbol{u}_t|^2}} \leq \frac{2\|\varepsilon_t\|_2}{\lambda_1|\boldsymbol{v}_1^\top\boldsymbol{u}_t|}\cdot\tan\theta(\boldsymbol{v}_1,\boldsymbol{u}_t) \leq 0.05\tan\theta(\boldsymbol{v}_1,\boldsymbol{u}_t). \tag{21}$$

where the last inequality is due to the condition that $\|\varepsilon_t\|_2 \leq \frac{\lambda_1|\boldsymbol{v}_1^\top\boldsymbol{u}_0|}{40} \leq \frac{\lambda_1|\boldsymbol{v}_1^\top\boldsymbol{u}_t|}{40}$. On the other hand, if $|\boldsymbol{v}_1^\top\boldsymbol{u}_t| > 0.5$ the following holds:

$$\beta = \frac{\|\mathbf{V}^\top\varepsilon_t\|_2}{\lambda_1|\boldsymbol{v}_1^\top\boldsymbol{u}_t|^2} \leq \frac{4\|\varepsilon_t\|_2}{\lambda_1} \leq \frac{4\tilde{\epsilon}_t}{\lambda_1}. \tag{22}$$

Combining Eq. (20,21,22) we obtain $\tan\theta(\boldsymbol{v}_1,\boldsymbol{u}_{t+1}) \leq 0.8\tan\theta(\boldsymbol{v}_1,\boldsymbol{u}_t) + 8\tilde{\epsilon}_t/\lambda_1$.

We next prove the first part of Eq. (9), namely that $\max_{j\neq 1}\lambda_j|\boldsymbol{v}_j^\top\boldsymbol{u}_{t+1}| \leq 0.5\lambda_1|\boldsymbol{v}_1^\top\boldsymbol{u}_{t+1}|$. For those $j$ with $\lambda_j = 0$ the bound trivially holds. So we consider only $j > 1$ with $\lambda_j > 0$. We then have that

$$\frac{\lambda_1|\boldsymbol{v}_1^\top\boldsymbol{u}_{t+1}|}{\lambda_j|\boldsymbol{v}_j^\top\boldsymbol{u}_{t+1}|} = \frac{\lambda_1|\boldsymbol{v}_1^\top[\mathbf{T}(\mathbf{I},\boldsymbol{u}_t,\boldsymbol{u}_t)+\varepsilon_t]|}{\lambda_j|\boldsymbol{v}_j^\top[\mathbf{T}(\mathbf{I},\boldsymbol{u}_t,\boldsymbol{u}_t)+\varepsilon_t]|} \geq \underbrace{\left(\frac{\lambda_1|\boldsymbol{v}_1^\top\boldsymbol{u}_t|}{\lambda_j|\boldsymbol{v}_j^\top\boldsymbol{u}_t|}\right)^2}_{\alpha'}\cdot\underbrace{\frac{1-|\boldsymbol{v}_1^\top\varepsilon_t|/(\lambda_1|\boldsymbol{v}_1^\top\boldsymbol{u}_t^2|)}{1+|\boldsymbol{v}_j^\top\varepsilon_t|/(\lambda_j|\boldsymbol{v}_j^\top\varepsilon_t|^2)}}_{\beta'}\cdot$$

By induction hypothesis $\alpha' \geq 4$. Applying conditions on $|\boldsymbol{v}_1^\top\varepsilon_t|$ we get $|\boldsymbol{v}_1^\top\varepsilon_t| \leq \frac{\lambda_1|\boldsymbol{v}_1^\top\boldsymbol{u}_0|^2}{4} \leq \frac{\lambda_1|\boldsymbol{v}_1^\top\boldsymbol{u}_t|^2}{4}$ and hence $|\boldsymbol{v}_1^\top\varepsilon_t|/(\lambda_1|\boldsymbol{v}_1^\top\boldsymbol{u}_t|^2) \leq 1/4$. On the other hand,

$$\left(\frac{\lambda_1|\boldsymbol{v}_1^\top\boldsymbol{u}_t|}{\lambda_j|\boldsymbol{v}_j^\top\boldsymbol{u}_t|}\right)^2\left[1+\frac{|\boldsymbol{v}_j^\top\varepsilon|}{\lambda_j|\boldsymbol{v}_j^\top\boldsymbol{u}_t|^2}\right]^{-1} = \left[\left(\frac{\lambda_j|\boldsymbol{v}_j^\top\boldsymbol{u}_t|}{\lambda_1|\boldsymbol{v}_1^\top\boldsymbol{u}_t|}\right)^2 + \frac{\lambda_j|\boldsymbol{v}_j^\top\varepsilon_t|}{\lambda_1^2|\boldsymbol{v}_1^\top\boldsymbol{u}_t|^2}\right]^{-1} \geq \left[\frac{1}{4}+\frac{|\boldsymbol{v}_j^\top\varepsilon_t|}{\lambda_1|\boldsymbol{v}_1^\top\boldsymbol{u}_t|^2}\right]^{-1}\cdot$$

Because $|\boldsymbol{v}_j^\top\varepsilon_t| \leq \frac{\lambda_1|\boldsymbol{v}_1^\top\boldsymbol{u}_0|^2}{8} \leq \frac{\lambda_1|\boldsymbol{v}_1^\top\boldsymbol{u}_t|^2}{8}$, the right-hand side of the above equation is lower bounded by $8/3$. Therefore, $\alpha'\beta' \geq \frac{8}{3}(1-\frac{1}{4}) \geq 2$.

The last part of this proof is devoted to showing Eq. (10). Under the condition that $\theta(\boldsymbol{v}_1,\boldsymbol{u}_t) \leq \pi/3$ we have that $\cos\theta(\boldsymbol{v}_1,\boldsymbol{u}_t) = |\boldsymbol{v}_1^\top\boldsymbol{u}_t| \geq 1/2$. Subsequently, for arbitrary $j > 1$ with $\lambda_j > 0$ the following holds:

$$\frac{|\boldsymbol{v}_j^\top\boldsymbol{u}_{t+1}|}{|\boldsymbol{v}_1^\top\boldsymbol{u}_{t+1}|} \leq \frac{\lambda_j|\boldsymbol{v}_j^\top\boldsymbol{u}_t|^2+|\boldsymbol{v}_j^\top\varepsilon_t|}{\lambda_1|\boldsymbol{v}_1^\top\boldsymbol{u}_t|^2-|\boldsymbol{v}_1^\top\varepsilon_t|} \leq \frac{|\boldsymbol{v}_j^\top\boldsymbol{u}_t|}{|\boldsymbol{v}_1^\top\boldsymbol{u}_t|}\cdot\underbrace{\frac{1}{2}\frac{1}{1-|\boldsymbol{v}_1^\top\varepsilon_t|/(\lambda_1|\boldsymbol{v}_1^\top\boldsymbol{u}_t|^2)}}_{\alpha} + \underbrace{\frac{|\boldsymbol{v}_j^\top\varepsilon_t|}{\lambda_1|\boldsymbol{v}_1^\top\boldsymbol{u}_t|^2-|\boldsymbol{v}_1^\top\varepsilon_t|}}_{\gamma}\cdot$$

Because $|\boldsymbol{v}_1^\top\boldsymbol{u}_t| \geq 1/2$ and $|\boldsymbol{v}_1^\top\varepsilon_t| \leq \frac{1}{2}\lambda_1|\boldsymbol{v}_1^\top\boldsymbol{u}_0|^2 \leq \frac{1}{2}\lambda_1|\boldsymbol{v}_1^\top\boldsymbol{u}_t|^2$, we have $\gamma \leq 8|\boldsymbol{v}_j^\top\varepsilon_t|/\lambda_1$ and hence

$$|\boldsymbol{v}_j^\top\boldsymbol{u}_{t+1}| \leq 0.8|\boldsymbol{v}_j^\top\boldsymbol{u}_t| + \frac{8|\boldsymbol{v}_j^\top\varepsilon_t|}{\lambda_1} \leq 0.8|\boldsymbol{v}_j^\top\boldsymbol{u}_t| + \frac{8\tilde{\epsilon}_t}{\lambda_1\sqrt{k}}.$$

$$\square$$

*Proof of Lemma C.3.* The first part of Eq. (12) is a simplified result of Lemma B.5 [4] in [1] because $\|\hat{\boldsymbol{v}}_i - \boldsymbol{v}_i\|_2 \leq \tan\theta(\hat{\boldsymbol{v}}_i, \boldsymbol{v}_i)$ when $\|\hat{\boldsymbol{v}}_i\|_2 = \|\boldsymbol{v}_i\|_2 = 1$ and $\theta < \pi/2$. The proofs are almost identical. So we focus on proving the second part of Eq. (12) here. Recall that $\boldsymbol{v}_j^\top \boldsymbol{v}_i = 0$ for all $j > i$. Subsequently,

$$\left| \sum_{i=1}^t \mathbf{E}_i(\boldsymbol{v}_j, \boldsymbol{u}, \boldsymbol{u}) \right| \leq \sum_{i=1}^t \hat{\lambda}_i |\boldsymbol{u}^\top \hat{\boldsymbol{v}}_i|^2 |\boldsymbol{v}_j^\top \hat{\boldsymbol{v}}_i| \leq \frac{C\epsilon}{\sqrt{k}} \sum_{i=1}^t \frac{\hat{\lambda}_i}{\lambda_i} |\boldsymbol{u}^\top \hat{\boldsymbol{v}}_i|^2.$$

Define $\hat{\boldsymbol{v}}_i^\perp = \hat{\boldsymbol{v}}_i - (\hat{\boldsymbol{v}}_i^\top \boldsymbol{v}_i)\boldsymbol{v}_i$ as the difference between $\hat{\boldsymbol{v}}_i$ and its projection on $\boldsymbol{v}_i$. It is then by definition that $\|\hat{\boldsymbol{v}}_i^\perp\|_2 = \|\hat{\boldsymbol{v}}_i\|_2 \sin\theta(\hat{\boldsymbol{v}}_i, \boldsymbol{v}_i) \leq \tan\theta(\hat{\boldsymbol{v}}_i, \boldsymbol{v}_i)$. Subsequently,

$$\sum_{i=1}^t \frac{\hat{\lambda}_i}{\lambda_i} |\boldsymbol{u}^\top \hat{\boldsymbol{v}}_i|^2 \leq \sum_{i=1}^t \left( 1 + \frac{|\hat{\lambda}_i - \lambda_i|}{\lambda_i} \right) |\boldsymbol{u}^\top \hat{\boldsymbol{v}}_i|^2 \leq \frac{C\epsilon}{\lambda_{\min}} + \sum_{i=1}^t \left( |\boldsymbol{u}^\top \boldsymbol{v}_i|^2 + |\boldsymbol{u}^\top \hat{\boldsymbol{v}}_i^\perp|^2 \right)$$

$$\leq \frac{C\epsilon}{\lambda_{\min}} + k\|\hat{\boldsymbol{v}}_i^\perp\|_2^2 + \sum_{i=1}^t |\boldsymbol{u}^\top \boldsymbol{v}_i|^2 \leq \underbrace{\frac{C\epsilon}{\lambda_{\min}} + \frac{C^2 k\epsilon^2}{\lambda_{\min}^2}}_{a} + \sum_{i=1}^t |\boldsymbol{u}^\top \boldsymbol{v}_i|^2.$$

For arbitrary $\delta \in (0,1)$, set $C' \leq \min\{\frac{\delta}{2C^2}, \frac{\delta}{2C^3}\}$ and $\epsilon \leq C'\lambda_{\min}/\sqrt{k}$ we have that $a \leq \delta/C$ and hence the second part of Eq. (12) holds with $C'' = C$. □

## D    Proof of results for streaming robust tensor power method

*Proof of Theorem 3.1.* First, note that if $\boldsymbol{x}_1, \cdots, \boldsymbol{x}_n \overset{i.i.d.}{\sim} P$, $P \in \mathcal{SG}_D(\sigma)$ then the distribution of the sample mean $\bar{\boldsymbol{x}} = \frac{1}{n}\sum_{i=1}^n \boldsymbol{x}_i$ belongs to $\mathcal{SG}_D(\sigma/\sqrt{n})$. To see this, fix any $\boldsymbol{a} \in \mathbb{R}^D$ and one can show that

$$\mathbb{E}\left[ \exp(\boldsymbol{a}^\top \bar{\boldsymbol{x}}) \right] = \prod_{i=1}^n \mathbb{E}\left[ \exp(\boldsymbol{a}^\top \boldsymbol{x}_i/n) \right] \leq \prod_{i=1}^n \exp(\|\boldsymbol{a}\|_2^2 \sigma^2/n^2) = \exp(\|\boldsymbol{a}\|_2^2 \sigma^2/n),$$

where the second inequality is due to the fact that $\boldsymbol{x}_i \in \mathcal{SG}_D(\sigma)$ and $\|\boldsymbol{a}/n\|_2^2 = \|\boldsymbol{a}\|_2^2/n^2$.

Under Assumptions 3.1, 3.2 and using the the above arguments, we know that

$$\text{vec}(\boldsymbol{\Delta}_T) = \text{vec}\left[ \frac{1}{n}\sum_{i=1}^n \boldsymbol{x}_i^{\otimes 3} - \mathbf{T} \right] \in \mathcal{SG}_{d^3}(\sigma/n)$$

Now fix $\boldsymbol{v}_i, \boldsymbol{u}_t \in \mathbb{R}^d$ with unit $L_2$ norms. Applying Lemma F.6 with respect to $\boldsymbol{\Sigma} = \text{vec}(\boldsymbol{v}_i \otimes \boldsymbol{u}_t \otimes \boldsymbol{u}_t)\text{vec}(\boldsymbol{v}_i \otimes \boldsymbol{u}_t \otimes \boldsymbol{u}_t)^\top$ we obtain that

$$\Pr\left[ \left|\boldsymbol{\Delta}_T(\boldsymbol{v}_i, \boldsymbol{u}_t, \boldsymbol{u}_t)\right|^2 > (1 + 2\sqrt{t} + t)\frac{\sigma^2}{n} \right] \leq e^{-t}, \quad \forall t > 0. \tag{23}$$

Subsequently, with overwhelming probability (e.g., $\geq 1 - n^{-10}$) we have that

$$\|\boldsymbol{\Delta}_T(\mathbf{I}, \boldsymbol{u}_t, \boldsymbol{u}_t)\|_2 = \widetilde{O}\left( \sigma\sqrt{\frac{d}{n}} \right), \quad |\boldsymbol{\Delta}_T(\boldsymbol{v}_i, \boldsymbol{u}_t, \boldsymbol{u}_t)| = \widetilde{O}\left( \sigma\sqrt{\frac{1}{n}} \right).$$

Finally, with

$$n = \widetilde{\Omega}\left( \min\left\{ \frac{\sigma^2 d}{\epsilon^2}, \frac{\sigma^2 d^2}{\lambda_{\min}^2} \right\} \right)$$

the conditions in Eq. (4) are satisfied with overwhelming probability and hence the error bounds on $|\lambda_i - \hat{\lambda}_{\pi(i)}|$ and $\|\boldsymbol{v}_i - \hat{\boldsymbol{v}}_{\pi(i)}\|_2$. □

# E  Proofs of utility results for differentially private tensor decomposition

Before proving Theorem 4.2, we first present a lemma that upper bounds $\|u_t\|_\infty$ when the components $\mathbf{V} \in \mathbb{R}^{d \times k}$ is incoherent (Assumption 4.1) and Gaussian noise across power updates is added.

**Lemma E.1.** *Suppose* $\mathbf{T} = \sum_{i=1}^k \lambda_i v_i^{\otimes 3}$ *and* $\mathbf{V} = (v_1, \cdots, v_k)$ *satisfies Assumption 4.1 with coherence level* $\mu_0$. *Fix* $u \in \mathbb{R}^d$ *with* $\|u\|_2 = 1$ *and let* $\bar{u} = \mathbf{T}(\mathbf{I}, u, u) + \sigma \cdot z$, *where* $z \sim \mathcal{N}(\mathbf{0}, \mathbf{I}_{d \times d})$ *are zero-mean independently distributed Gaussian random variables. We then have that*

$$\frac{\|\bar{u}\|_\infty}{\|\bar{u}\|_2} = O\left(\sqrt{\frac{\mu_0 k \log d}{d}}\right).$$

*with overwhelming probability.*

*Proof.* We prove this lemma by showing an upper bound for $\|\bar{u}\|_\infty$ and a lower bound on $\|\bar{u}\|_2$, both with overwhelming probabilities. For the infinity-norm upper bound, we consider the following decomposition via triangle inequality:

$$\|\bar{u}\|_\infty \le \|\tilde{u}\|_\infty + \sigma \|z\|_\infty,$$

where $\tilde{u} = \mathbf{T}(\mathbf{I}, u, u)$ and $z \sim \mathcal{N}(\mathbf{0}, \mathbf{I}_{d \times d})$. By definition,

$$\|\tilde{u}\|_\infty = \left\| \sum_{i=1}^k \lambda_i |u^\top v_i|^2 v_i \right\|_\infty = \max_{1 \le j \le d} \left| \lambda^\top (\mathbf{V}^\top e_j) \right|,$$

where $\lambda$ is a $k$-dimensional vector defined as $\lambda = (\lambda_1 |u^\top v_1|^2, \cdots, \lambda_k |u^\top v_k|^2)$. By Cauchy-Schwarts inequality, we have that

$$\|\tilde{u}\|_\infty = \max_{1 \le j \le d} \left| \lambda^\top (\mathbf{V}^\top e_j) \right| \le \|\lambda\|_2 \cdot \max_{1 \le j \le d} \|\mathbf{V}^\top e_j\|_2 \le \sqrt{\frac{\mu_0 k}{d} \left( \sum_{i=1}^k \lambda_i^2 |u^\top v_i|^4 \right)},$$

where the last inequality is due to the condition that $\mathbf{V}$ is incoherent with coherence level $\mu_0$. In addition, $\|z\|_\infty = O(\sqrt{\log d})$ with overwhelming probability, by applying Lemma F.3. As a result,

$$\|\bar{u}\|_\infty \le \sqrt{\frac{2k\mu_0}{d} \left( \sum_{i=1}^k \lambda_i^2 |u^\top v_i|^4 \right)} + O(\sigma \sqrt{\log d}). \tag{24}$$

We next lower bound the denominator term $\|\bar{u}\|_2$. By definition, $\bar{u}$ follows a multi-variate Gaussian distribution with mean $\tilde{u}$ and co-variance $\sigma^2 \mathbf{I}_{d \times d}$. Applying Lemma F.5 with $\mu = \|\tilde{u}\|_2^2 / \sigma^2$ and $t = O(\log d)$ we have that $\|\bar{u}\|_2^2 = \Omega(\|\tilde{u}\|_2^2 + \sigma^2 d)$ with overwhelming probability. Note also that

$$\|\tilde{u}\|_2^2 = \left\| \sum_{i=1}^k \lambda_i |u^\top v_i|^2 v_i \right\|_2^2 = \sum_{i=1}^k \lambda_i^2 |u^\top v_i|^4$$

because $\{v_i\}_{i=1}^k$ are orthonormal vectors. Consequently,

$$\|\bar{u}\|_2^2 = \Omega\left( \sqrt{\sigma^2 d + \sum_{i=1}^k \lambda_i^2 |u^\top v_i|^4} \right). \tag{25}$$

Combining Eqs. (24,25) we obtain

$$\frac{\|\bar{u}\|_\infty}{\|\bar{u}\|_2} \le \frac{\sqrt{\frac{2\mu_0 k}{d} \sum_{i=1}^k \lambda_i^2 |u^\top v_i|^4} + O(\sigma \sqrt{\log d})}{\Omega\left( \sqrt{\sigma^2 d + \sum_{i=1}^k \lambda_i^2 |u^\top v_i|^4} \right)} \le O\left( \sqrt{\frac{\mu_0 k}{d}} \right) + O\left( \sqrt{\frac{\log d}{d}} \right) = O\left( \sqrt{\frac{\mu_0 k \log d}{d}} \right).$$

$\square$

We are now ready to prove Theorem 4.2.

*Proof of Theorem 4.2.* Applying Lemma E.1 we can with overwhelming probability upper bound the per-coordinate standard deviation of Gaussian noise calibrated in Algorithm 3:

$$\max_{\substack{0 \leq t \leq T \\ 1 \leq \tau \leq L}} \left\{ \nu \|\boldsymbol{u}_t^{(\tau)}\|_\infty^2, \nu \|\boldsymbol{u}_t^{(\tau)}\|_\infty^3 \right\} \leq O\left( \frac{\sqrt{K} \cdot \log(1/\delta)}{\varepsilon} \cdot \frac{\mu_0 k \log d}{d} \right),$$

where $K = kL(T+1) = \widetilde{O}(k^2 \log(\lambda_{\max}d))$. Let $\boldsymbol{\epsilon}_t^{(\tau)} = \mathbf{E}(\mathbf{I}, \boldsymbol{u}_t^{(\tau)}, \boldsymbol{u}_t^{(\tau)}) = \sigma_t^{(\tau)} \cdot \boldsymbol{z}$ be the noise vector calibrated, where $\sigma_t^{(\tau)} = \nu \|\boldsymbol{u}_t^{(\tau)}\|_\infty^2$. We then have that with overwhelming probability,

$$\|\boldsymbol{\epsilon}_t^{(\tau)}\|_2 = O\left( \frac{\mu_0 k^2 \log(\lambda_{\max}d/\delta)}{\varepsilon\sqrt{d}} \right) \qquad \text{and} \qquad |\boldsymbol{v}_1^\top \boldsymbol{\epsilon}_t^{(\tau)}| = O\left( \frac{\mu_0 k^2 \log(\lambda_{\max}d/\delta)}{\varepsilon d} \right).$$

Equating the upper bound for $|\boldsymbol{v}_1^\top \boldsymbol{\epsilon}_t^{(\tau)}|$ with $O(\lambda_{\min}/d)$ we obtain the desired privacy level condition:

$$\varepsilon = \Omega\left( \frac{\mu_0 k^2 \log(\lambda_{\max}d/\delta)}{\lambda_{\min}} \right).$$

It can also be easily verified that all noise conditions in Theorem 2.2 are satisfied with above lower bound condition on $\varepsilon$. □

# F    Technical lemmas

## F.1    Tail inequalities

**Lemma F.1** (Tail bound of Lipschitz function of Gaussian random variables, [8]). *Suppose $\boldsymbol{x} \sim \mathcal{N}_d(0, \sigma^2 \mathbf{I}_{d \times d})$ are d-dimensional independent Gaussian random variables and let $f : \mathbb{R}^d \to \mathbb{R}$ be an L-Lipschitz function; that is, $|f(\boldsymbol{x}) - f(\boldsymbol{y})| \leq L\|\boldsymbol{x} - \boldsymbol{y}\|_2$ for all $\boldsymbol{x}, \boldsymbol{y} \in \mathbb{R}^d$. Suppose $\mu = \mathbb{E}_{\boldsymbol{x}}[f(\boldsymbol{x})]$. Then for all $t > 0$, we have that*

$$\Pr\left[ \big| f(\boldsymbol{x}) - \mu \big| \geq t \right] \leq 2e^{-t^2/(2\sigma^2 L^2)}.$$

**Lemma F.2** (Bounds on maximum of Gaussian random variables, [19]). *Suppose $X_1, \cdots, X_n \overset{i.i.d.}{\sim} \mathcal{N}(0, \sigma^2)$ and let $Y = \max_{1 \leq i \leq n} X_i$. We then have that*

$$\frac{\sigma}{\sqrt{\pi \ln 2}} \sqrt{\ln n} \leq \mathbb{E}[Y] \leq \sigma\sqrt{2}\sqrt{\ln n}.$$

**Lemma F.3** (Bounds on maximum absolute values of Gaussian random variables; Theorem 3.12, [23]). *Suppose $X_1, \cdots, X_n \overset{i.i.d.}{\sim} \mathcal{N}(0, \sigma^2)$ and let $Y = \max_{1 \leq i \leq n} |X_i|$. We then have that*

$$\Pr\left[ Y \geq \sigma\sqrt{2\ln n} + \sigma\sqrt{2t} \right] \leq e^{-t}, \quad \forall t > 0.$$

**Lemma F.4** (Bounds on Chi-square random variables, [22]). *Suppose $X \sim \chi_k^2$; that is, $X = \sum_{j=1}^k Y_j^2$ for i.i.d. standard Normal random variables $Y_1, \cdots, Y_k$. We then have that $\forall t > 0$,*

$$\Pr\left[ X \geq k + 2\sqrt{kt} + 2t \right] \leq e^{-t}, \qquad \Pr\left[ X \leq k - 2\sqrt{kt} \right] \leq e^{-t}.$$

**Lemma F.5** (Bounds on non-central Chi-square random variables, [7]). *Suppose $X \sim \chi_k^2(\mu)$; that is, $X = \sum_{j=1}^k Y_j^2$ for independent Normal random variables $Y_1, \cdots, Y_k$ distributed as $Y_j \sim \mathcal{N}(\mu_j, 1)$, $\sum_j \mu_j = \mu$. We then have that*

$$\begin{aligned} \Pr\left[ X \geq (k + \mu) + 2\sqrt{(k + 2\mu)t} + 2t \right] &\leq e^{-t}, \\ \Pr\left[ X \leq (k + \mu) - 2\sqrt{(k + 2\mu)t} \right] &\leq e^{-t}. \end{aligned}$$

**Lemma F.6** (Bounds on quadratic forms of sub-Gaussian random variables, [15])**.** *Suppose* $X \sim \mathcal{SG}_D(\sigma)$ *and let* $\Sigma \in \mathbb{R}^{D \times D}$ *be a positive semidefinite matrix. Then for all* $t > 0$ *we have that*

$$\Pr\left[X^\top \Sigma X > \sigma^2 \left(\text{tr}(\Sigma) + 2\sqrt{\text{tr}(\Sigma^2)t} + 2\|\Sigma\|t\right)\right] \leq e^{-t}.$$

## F.2 Matrix perturbation lemmas

**Lemma F.7** (Weyl's theorem; Theorem 4.11, p. 204 in [26])**.** *Let* $\mathbf{A}, \mathbf{E}$ *be given* $m \times n$ *matrices with* $m \geq n$. *Then*

$$\max_{i \in [n]} \left|\sigma_i(\mathbf{A} + \mathbf{E}) - \sigma_i(\mathbf{A})\right| \leq \|\mathbf{E}\|_2.$$

**Lemma F.8** (Wedin's theorem; Theorem 4.4, pp. 262 in [26])**.** *Let* $\mathbf{A}, \mathbf{E} \in \mathbb{R}^{m \times n}$ *be given matrices with* $m \geq n$. *Let* $\mathbf{A}$ *have the following singular value decomposition*

$$\begin{bmatrix} \mathbf{U}_1^\top \\ \mathbf{U}_2^\top \\ \mathbf{U}_3^\top \end{bmatrix} \mathbf{A} \begin{bmatrix} \mathbf{V}_1 & \mathbf{V}_2 \end{bmatrix} = \begin{bmatrix} \mathbf{\Sigma}_1 & \mathbf{0} \\ \mathbf{0} & \mathbf{\Sigma}_2 \\ \mathbf{0} & \mathbf{0} \end{bmatrix},$$

*where* $\mathbf{U}_1, \mathbf{U}_2, \mathbf{U}_3, \mathbf{V}_1, \mathbf{V}_2$ *have orthonormal columns and* $\mathbf{\Sigma}_1$ *and* $\mathbf{\Sigma}_2$ *are diagonal matrices. Let* $\widetilde{\mathbf{A}} = \mathbf{A} + \mathbf{E}$ *be a perturbed version of* $\mathbf{A}$ *and* $(\widetilde{\mathbf{U}}_1, \widetilde{\mathbf{U}}_2, \widetilde{\mathbf{U}}_3, \widetilde{\mathbf{V}}_1, \widetilde{\mathbf{V}}_2, \widetilde{\mathbf{\Sigma}}_1, \widetilde{\mathbf{\Sigma}}_2)$ *be analogous singular value decomposition of* $\widetilde{\mathbf{A}}$. *Let* $\mathbf{\Phi}$ *be the matrix of canonical angles between* $\text{Range}(\mathbf{U}_1)$ *and* $\text{Range}(\widetilde{\mathbf{U}}_1)$ *and* $\mathbf{\Theta}$ *be the matrix of canonical angles between* $\text{Range}(\mathbf{V}_1)$ *and* $\text{Range}(\widetilde{\mathbf{V}}_1)$. *If there exists* $\alpha, \delta > 0$ *such that*

$$\min_i \sigma_i(\widetilde{\mathbf{\Sigma}}_1) \geq \alpha + \delta \quad \text{and} \quad \max_i \sigma_i(\mathbf{\Sigma}_2) \leq \alpha,$$

*then*

$$\max\{\|\mathbf{U}_1\mathbf{U}_1^\top - \widetilde{\mathbf{U}}_1\widetilde{\mathbf{U}}_1^\top\|_2, \|\mathbf{U}_1\mathbf{U}_1^\top - \widetilde{\mathbf{V}}_1\widetilde{\mathbf{V}}_1^\top\|_2\} = \max\{\|\sin\mathbf{\Phi}\|_2, \|\sin\mathbf{\Theta}\|_2\} \leq \frac{\|\mathbf{E}\|_2}{\delta}.$$

## F.3 Lemmas on random tensors

**Lemma F.9** (Spectral norm bound of random tensors, [27])**.** *Suppose* $\mathbf{X}$ *is a* $p$th *order tensor with dimensions* $d_1, \cdots, d_p$ *and each element of* $\mathbf{X}$ *is sampled i.i.d. from Gaussian distribution* $\mathcal{N}(0, \sigma^2)$. *Then the following upper bound on* $\|\mathbf{X}\|_{\text{op}}$ *holds with probability at least* $(1 - \delta)$:

$$\|\mathbf{X}\|_{\text{op}} \leq \sqrt{8\sigma^2\left(\left(\sum_{k=1}^p d_p\right)\ln(2K/K_0) + \ln(2/\delta)\right)},$$

*where* $K_0 = \ln(3/2)$.

## Footnotes

[3] For notational simplicity, let $\tan \theta(\boldsymbol{v}_1, \boldsymbol{u}_{-1}) = \infty$.

[4]Except that we operate under a $k < d$ regime, which adds no difficulty to the proof.