[Reviews · NeurIPS 2016]

Reviewer 1

Summary

NOTE TO AUTHORS AND TPC: I was asked to provide an additional review for this paper after the rebuttal period, but did not read the other reviews or rebuttal prior to writing this review. I received some assistance in this review from a graduate student who also read the paper. This is a "light" review. I did not have an opportunity to verify the proofs or verify correctness. SUMMARY: This paper proposes a new method for online tensor decomposition based on the power iteration and in particular on its robustness to noise. They then (perhaps inspired by Hardt and Price) suggest that this method could give effective differentially private algorithms for the power iteration, provided the noise level is sufficiently low (or \epsilon is sufficiently large).

Qualitative Assessment

STRENGTHS: - The problem addressed by the authors is important - The proposed algorithm is relatively novel and seems effective - While I didn't verify the proofs, the arguments appear correct. WEAKNESSES: - I found the application to differential privacy unconvincing (see comments below) - Experimental validation was a bit light and felt preliminary RECOMMENDATION: I think this paper should be accepted into the NIPS program on the basis of the online algorithm and analysis. However, I think the application to differential privacy, without experimental validation, should be omitted from the main paper in favor of the preliminary experimental evidence of the tensor method. The results on privacy appear too preliminary to appear in a "conference of record" like NIPS. TECHNICAL COMMENTS: 1) Section 1.2: the dimensions of the projection matrices are written as $A_i \in \mathbb{R}^{m_i \times d_i}$. I think this should be $A_i \in \mathbb{R}^{d_i \times m_i}$, otherwise you cannot project a tensor $T \in \mathbb{R}^{d_1 \times d_2 \times \ldots d_p}$ on those matrices. But maybe I am wrong about this... 2) The neighborhood condition in Definition 3.2 for differential privacy seems a bit odd in the context of topic modeling. In that setting, two tensors/databases would be neighbors if one document is different, which could induce a change of something like $\sqrt{2}$ (if there is no normalization, so I found this a bit confusing. This makes me think the application of the method to differential privacy feels a bit preliminary (at best) or naive (at worst): even if a method is robust to noise, a semantically meaningful privacy model may not be immediate. This $\sqrt{2}$ is less than the $\sqrt{6}$ suggested by the authors, which may make things better? 3) A major concern I have about the differential privacy claims in this paper is with regards to the noise level in the algorithm. For moderate values of $L$, $R$, and $K$, and small $\epsilon = 1$, the noise level will be quite high. The utility theorem provided by the author requires a lower bound on $\epsilon$ to make the noise level sufficiently low, but since everything is in "big-O" notation, it is quite possible that the algorithm may not work at all for reasonable parameter values. A similar problem exists with the Hardt-Price method for differential privacy (see a recent ICASSP paper by Imtiaz and Sarwate or an ArXiV preprint by Sheffet). For example, setting L=R=100 and K=10, \epsilon = 1, \delta = 0.01 then the noise variance is of the order of 4 x 10^4. Of course, to get differentially private machine learning methods to work in practice, one either needs large sample size or to choose larger $\epsilon$, even $\epsilon \gg 1$. Having any sense of reasonable values of $\epsilon$ for a reasonable problem size (e.g. in topic modeling) would do a lot towards justifying the privacy application. 4) Privacy-preserving eigenvector computation is pretty related to private PCA, so one would expect that the authors would have considered some of the approaches in that literature. What about (\epsilon,0) methods such as the exponential mechanism (Chaudhuri et al., Kapralov and Talwar), Laplace noise (the (\epsilon,0) version in Hardt-Price), or Wishart noise (Sheffet 2015, Jiang et al. 2016, Imtiaz and Sarwate 2016)? 5) It's not clear how to use the private algorithm given the utility bound as stated. Running the algorithm is easy: providing $\epsilon$ and $\delta$ gives a private version -- but since the $\lambda$'s are unknown, verifying if the lower bound on $\epsilon$ holds may not be possible: so while I get a differentially private output, I will not know if it is useful or not. I'm not quite sure how to fix this, but perhaps a direct connection/reduction to Assumption 2.2 as a function of $\epsilon$ could give a weaker but more interpretable result. 6) Overall, given 2)-5) I think the differential privacy application is a bit too "half-baked" at the present time and I would encourage the authors to think through it more clearly. The online algorithm and robustness is significantly interesting and novel on its own. The experimental results in the appendix would be better in the main paper. 7) Given the motivation by topic modeling and so on, I would have expected at least an experiment on one real data set, but all results are on synthetic data sets. One problem with synthetic problems versus real data (which one sees in PCA as well) is that synthetic examples often have a "jump" or eigenvalue gap in the spectrum that may not be observed in real data. While verifying the conditions for exact recovery is interesting within the narrow confines of theory, experiments are an opportunity to show that the method actually works in settings where the restrictive theoretical assumptions do not hold. I would encourage the authors to include at least one such example in future extended versions of this work.

Confidence in this Review

3-Expert (read the paper in detail, know the area, quite certain of my opinion)


Reviewer 2

Summary

The paper presents two tensor decomposition algorithms for empirical moment tensors. One is an online version of the well-known power method with reduced memory requirements over the batch case. The other is a differentially private version that can privately recover the leading eigenvector and eigenvalue of a symmetric moment tensor. The analysis of both algorithms relies on an improved analysis of the noise tolerance of the power method.

Qualitative Assessment

The streaming tensor decomposition presented in the paper can be of great practical utility. On the other hand, the differentially private decomposition seems still too rudimentary (is only able to recover the top component of the tensor) to have any kind of practical application. The paper is well written and most of the important concepts and ideas are explained with enough detail. However, there are a couple of points which need to be clarified: a) The comparison with [25] (line 61) states that this paper already presents an approach to tensor decomposition which can tolerate noise up to O(d^-1/2) but requires the computation of rank-1 tensor approximations. But isn’t the computation of a rank-1 approximation exactly what one iteration of the power method does? How different is your analysis from an adaptation of [25] to the case where the rank-1 approximations are recovered up to some accuracy? b) What is exactly the role of the incoherence assumption on the utility guarantees of the differentially private algorithm? Is this condition necessary to achieve any kind of recovery under privacy constraints, or is it just assumed to make the analysis simpler? c) Would assumption 2.2 on the sub-gaussianity of the centered third order moment of x follow from the subgaussianity of x? Is it worth making the distinction and introduce this seemingly less standard assumption?

Confidence in this Review

2-Confident (read it all; understood it all reasonably well)


Reviewer 3

Summary

This paper gives a noisy power method for differentially private decomposition of a third-order tensor. The method is a natural extension of the noisy power method of Hardt and Roth for matrix decomposition. As in Hardt and Roth, they give an analysis in terms of the coherence of the tensor.

Qualitative Assessment

I am rather confused about the way the utility result is stated. (eps, delta)-differenital privacy is only interested when eps is smaller than 1 and delta is much smaller than 1/n, so typically we fix these parameters, and then state the utility in terms of them. It doesn't make sense to give bounds on eps in terms of other parameters, because I would never be satisfied with the privacy of an algorithm that uses eps bigger than 1. After some work, I think that the utility theorem could still be meaningful, but it's very confusingly written. If the coherence is a constant, delta is set appropriately, and the rank is a constant, then the condition basically says that eps can be taken to be some reasonable parameter whenever log(lambda_max d n) / lambda_min smaller than 1. The numerator is going to be bigger than 1 in most reasonable settings, so we only get reasonable privacy if lambda_min greater than 1. But this confuses me since you'd think that multipling the tensor by a scalar shouldn't fundamentally change the utility (it changes the scale of the "true answer" but the noise for privacy should be scaled in the same way), so it's very confusing that I get better privacy if I multiply by a constant. This problem might go away if the utlity were stated as a function of the privacy parameters (and other parameters), rather than fixing the utility and stating privacy as a function of just the other parameters. The bottom line is I cannot tell in this form.

Confidence in this Review

2-Confident (read it all; understood it all reasonably well)


Reviewer 4

Summary

This paper presents improvements to the Robust Tensor Power Method that make it better suited to practical applications. For the common setting where the tensor to be decomposed is the empirical moments of some random variable $x$ and we have access to IID samples of $x$, their ''Online Robust Tensor Power Method'' has time and space complexity linear in the dimension $d$ and the rank $k$ of the tensor to be decomposed. Compared to the basic TPM, the new method is linearly convergent, whereas the original has quadratic convergence. The authors also propose a differentially private version of the robust TPM and show that 1) it is ($\eps$, $\delta$)-differentially private and 2) When the privacy parameter $\eps$ is sufficiently large (poly(rank, incoherence, 1/\lambda_{min})) , the method recovers the principal eigenvector. Finally the authors provide a more fine-grained analysis of the robust tensor power method in the setting where the tensor to be decomposed has additive noise. This shows that when the noise tensor $\Delta$ has special structure, e.g. is subgaussian, recovery is possible even when the operator norm of $\Delta$ is larger than the best uniform upper bound for the operator norm under which recovery is possible. This implies that for some noisy settings the new online and differentially private algorithms have compelling performance in spite of their having worse dependence on precision parameters than the vanilla TPM.

Qualitative Assessment

This paper is very nicely written. As tensor-based algorithms are becoming increasingly common, at least in theory, making these methods practical -- especially in terms of time and memory complexity -- seems important. To this end I believe that the Online Tensor Power Method (Algorithm 2) is the most important contribution of the paper. The population moment setting for which this algorithm applies seems to capture many applications. Theorem 4.2 seems important as it sheds light on which which realistic conditions we should expect the online tensor power method have reasonable performance. It interesting to see experimentally (Figure 1.) that some of these thresholds seem to be sharp. I'm less convinced of the potential impact of the differentially private TPM (Algorithm 3) -- this seems like a fairly niche topic and it's not clear that the analysis itself is substantially new or of broad interest. My main criticism of the paper as a whole is that many of the results on their own feel somewhat incremental, but I think that combined they tell a coherent story and form a useful body of work, and for this reason I am in favor of accepting. It would have been nice to include at least one end-to-end analysis of streaming and/or differentially private recovery for an application like community detection -- this is mentioned in the conclusion as a future direction. Given that this paper focuses on improvements to the TPM for practical settings, it would have been nice to have seen some experiments. I would be curious to see how the online TPM performs in terms of wall time for medium-sized problems. Misc comments: * Algorithm 2 uses i as index for two different things - this is a little confusing. * It seems like the RHS in Definition 3.2 should scale with T-T'. * Algorithm 4, line 3: Seems like the range should be 1,...,L -- I don't see where $\Tau$ is defined. * Line 171: Should be [1, d/k]?

Confidence in this Review

2-Confident (read it all; understood it all reasonably well)


Reviewer 5

Summary

The paper provides a method for differentially private tensor decomposition based on some extensions of the power method for tensors. The method is relatively time and space efficient, and also noise resilient. The variants of the power method proposed in this paper enables memory efficient streaming tensor decomposition. Under low-rank and sub-gaussianity of the noise assumptions, it is shown that the proposed method learns the decomposition within $1/sqrt{d}$ accuracy, where $d$ is the dimension of the tensor, using $\tilde{O}(kd^2)$ samples, where $k$ is the rank of the tensor, and $\tilde{O}(kd)$ memory. The paper also gives the first specialized treatment of tensor decomposition under differential privacy. The authors give a differentially private version of the power method algorithm based using Gaussian noise perturbation. Under incoherence assumption, accuracy guarantees are derived for the resulting differentially private decomposition.

Qualitative Assessment

The paper provides fairly interesting results on tensor decomposition. It is also the first to study this problem under differential privacy. The results are new and non-trivial. However, the authors have not motivated the use of differential privacy well enough for this task. The way Theorem 3.1 is stated is a bit strange. Normally, one would fix $\epsilon$ and $\delta$, and derive bounds on the error in terms of $\epsilon, \delta$ (as well as the rest of the model parameters), but here a lower bound is first set on $\epsilon$ (making it unreasonably large for any meaningful privacy) and then the error bounds are free of $\epsilon$. I think the statement of the theorem needs to be revised.

Confidence in this Review

2-Confident (read it all; understood it all reasonably well)


Reviewer 6

Summary

This paper mainly considered the tensor decomposition problem from the view of memory efficiency and differential privacy. Then authors proposed two algorithms based on a careful perturbation analysis, in order to solve the corresponding problems. One is an online tensor power method which has a linear memory requirement, and the other one is a noise calibrated tensor power method with efficient privacy guarantees.

Qualitative Assessment

Different with some notable existing results on tensor decomposition, authors mainly considered the problem from practical perspectives (i.e. computation level and privacy level), and obtained novel and useful results, which in my personal view is the main contribution of this paper. I believe that the two algorithms proposed in the paper, especially the first online algorithm, may have a promising application in real world's problems. However, there are mainly two points I feel some confused: 1. For the first online algorithm, though it calculates the approximate eigenvalues and eigenvectors in an online way, I feel that it looks more like an offline algorithm. Once we have O(nkR) samples, then we can obtain the approximations. However, from the classical online learning's perspective, now, another stream of samples come, then how to update the existing approximations and obtain more accurate approximations with theoretical guarantees? 2. To the best of my knowledge, this paper seems to be the first paper about DP tensor decomposition, which is quite interesting. The main point I feel a little concerned is the definition of neighboring tensors. Just consider the matrix case, then the definition of neighboring tensors in the paper means there are only two different entries in two neighboring matrices. If we regard each row in the matrix as each user's data, then neighbor matrices actually represent there exists a different row. Even though consider the tensor symmetrization process, it seems quite different in these two concepts. So how to explain this difference?

Confidence in this Review

1-Less confident (might not have understood significant parts)


Reviewer 7

Summary

Note this review has been written after the first round of reviews upon request, and after the author rebuttal period. That said, the independent review was written without first viewing the author responses/other reviews. The paper delivers as advertised: online tensor decomposition with time/space linear in tensor dimension/rank; and the same approach to recovering the principal eigenvec while preserving differential privacy, apparently based on private (matrix) power iteration (Hardt & Price).

Qualitative Assessment

Overall I found the paper compelling: While a significant portion of the work is on DP, the results on online tensor decomposition appear to be of independent interest. And while the method of preserving DP may be based on existing proof techniques, its goal (tensor decomposition with DP) appears to be novel. Compared to existing techniques, the developed approach is superior in one or other way. For example, compared to [11] (also based on SGD) the presented approach is suitable for odd-order tensors; [24,14] less complex, making it more likely to enjoy impact; [25] does not require exact computation of rank-1 tensor approximation. The presentation is restricted to order-3 tensors. While it would appear that the results extend to higher orders, there is little discussion on the consequences to results and rates. Bounds being wp at least 0.9, hide dependence on the level of probability. Thm/Cor 2.1 do not involve epsilon. However more importantly perhaps, is that the main theorems are written in an unconventional way: setting the privacy parameters as functions of utility. The DP portions of the paper do appear to bear resemblance to existing private noisy matrix power methods. This reduces its novelty in terms of fundamental approach. A number of assumptions are placed on the setup, some seem reasonable, but others appear restrictive (Assumptions 2.2, 3.1 – low-rank and subgaussianity of the noise). Some assumptions appear challenging to verify in practice. Requiring ||Delta_T||_op to be upper-bounded by O(lambda_min/d) for example. Is it possible to run the DP algorithm (with privacy end-to-end that is?): wouldn't one need to know the lambda's in order to establish privacy? I believe the paper could have impact due to its analysis of online tensor decomposition, perhaps more so than the DP portions of the paper. Overall I found the presentation to be good, providing a solid coverage of related work with comparisons and relative dis/advantages discussed. Minor: The initial presentation is a little unconventional for a NIPS audience in tone. For example, "pervasive tool in the era of big data" is a little on the hyperbolic side.

Confidence in this Review

1-Less confident (might not have understood significant parts)


Reviewer 8

Summary

The paper considers applications of the tensor noisy power method for two problems: online tensor decomposition and differentially private tensor decomposition. The main goal in the former is to get memory-efficient results (linear in dimension, rather than cubic, as the vanilla tensor power method would give). The main goal in the latter is to get an epsilon-parameter in the differential privacy guarantee that is not polynomially dependent upon dimension. The main workhorse lemma in both is an improved analysis of the usual tensor noisy power method when the noise tensor satisfies additional properties other than a worst-case bound on the operator norm of the noise tensor.

Qualitative Assessment

In my opinion, the more fine-grained analysis of the noisy power method when the noise tensor is "special" is the most useful contribution of this paper. I imagine this might have other applications beyond online tensor decomposition. The issues with the paper that I have are with regard to the clarity of writing and helping interpret the bounds for the reader. More concretely, here are a few points: -- The paper really should be reorganized to put Theorem 4.2 first, and then say that this is more or less directly used for proving both the diff. privacy and the online decomposition result. I actually had trouble upon first reading understanding how Theorem 4.2 fits in. -- Similarly, proposition 4.1 seems out of place and confusing. This seems more appropriate to me as a remark to be placed in the appendix rather than a separate section. I could not even figure out if this was an original result or just stating some prior tensor-unfolding-based technique does not work. -- The differential privacy result is hard to interpret for me. Is an epsilon parameter depending logarithmically on d and linearly on k good? I imagine epsilon needs to be reasonably smaller than 1 for the result to be meaningful. Maybe a comparison to the matrix case would be useful for those who are not in the diff. privacy subfield? (The authors do argue that the Gaussian noise mechanism trivially will lead to polynomial dependency on d -- but on the other hand, why is logarithmic dependence interesting even?) -- The novelty in the proofs is not well-delineated. I am not that intimately familiar with Hardt-Price :the sketch of Theorem 4.2 says they "borrow" from the analysis there. Is the main borrowed piece using tan \theta as a potential function? Is there more? -- Some of the proofs need unpacking/polishing here and there: for example the formula after line 437/448 in the appendix needs a few words. Whenever alpha and beta are introduced in the proofs, it is desirable to properly define them rather than just mark them with curly brackets. -- Appendix E is quite badly written overall. I do not know what the "suLQ" framework is; f1,f2 are never defined; the proof of proposition 3.1 is quite vague overall. -- Finally, there are a few typos here and there (though nothing horrible). The most significant ones are in the statement of algorithm 2. In line 7, v--> kappa{v}; u --> \tilde{u} at the end of the line.

Confidence in this Review

2-Confident (read it all; understood it all reasonably well)